# Offline RL with Discrete Proxy Representations for Generalizability in POMDPs

**Pengjie Gu** [1,†]**, Xinyu Cai** [1,†]**, Dong Xing**[2]**, Xinrun Wang** [1,‡]**, Mengchen Zhao** [3,‡]**, Bo An**[1]

School of Computer Science and Engineering, Nanyang Technological University, Singapore[1]
College of Computer Science and Technology, Zhejiang University[3]
Noah's Ark Lab, Huawei[3]
{pengjie.gu, xinyu.cai, xinrun.wang, boan}@ntu.edu.sg,
dongxing@zju.edu.cn, zhaomengchen@huawei.com

## Abstract

Offline Reinforcement Learning (RL) has demonstrated promising results in various applications by learning policies from previously collected datasets, reducing the need for online exploration and interactions. However, real-world scenarios usually involve partial observability, which brings crucial challenges of the deployment of offline RL methods: i) the policy trained on data with full observability is not robust against the masked observations during execution, and ii) the information of which parts of observations are masked is usually unknown during training. In order to address these challenges, we present Offline RL with DiscrEte pRoxy representations (ORDER), a probabilistic framework which leverages novel state representations to improve the robustness against diverse masked observabilities. Specifically, we propose a discrete representation of the states and use a proxy representation to recover the states from masked partial observable trajectories. The training of ORDER can be compactly described as the following three steps. i) Learning the discrete state representations on data with full observations, ii) Training the decision module based on the discrete representations, and iii) Training the proxy discrete representations on the data with various partial observations, aligning with the discrete representations. We conduct extensive experiments to evaluate ORDER, showcasing its effectiveness in offline RL for diverse partially observable scenarios and highlighting the significance of discrete proxy representations in generalization performance. ORDER is a flexible framework to employ any offline RL algorithms and we hope that ORDER can pave the way for the deployment of RL policy against various partial observabilities in the real world.

## 1 Introduction

The past decade witnesses the remarkable success of Reinforcement Learning (RL) in various domains, such as Atari games [38], Go [43], and even nuclear fusion [10]. Offline RL, a particularly promising approach, focuses on learning effective policies from previously collected datasets, reducing the need for online exploration and interactions [34]. However, the various partial observabilities in the real world pose challenges when deploying the learned policy in practice. Examples include autonomous cars facing blocked or perturbed observations [11], drones navigating through cluttered environments with occlusions [51], and robotic manipulators handling objects with varying degrees of visibility [5]. As a result, agents trained on offline datasets must adapt their learned policies to cope with dynamic, uncertain, and partially observable environments, ensuring generalizability in real-world applications.

---

[1]† Equal contribution.
[2]‡ Corresponding authors.

37th Conference on Neural Information Processing Systems (NeurIPS 2023).

Our research targets the situations where agents, trained on offline datasets with full state information, encounter various partial observations in real-world deployments. For instance, a robot trained in a lab with complete data on obstacles and victims might face limited visibility due to occlusions, sensor noise, or environmental factors like smoke or varying light in real scenarios. This contrast between training and real-world conditions can hinder the system's generalizability and effectiveness.

There are two main challenges for the generalization against partial observabilities. i) During testing, the partial observability may differ substantially from the data on which the RL policy was trained, leading to significantly reduced performance. ii) The specific forms of partial observability remain unidentified during training, preventing the tailored training of agents for these observabilities. These challenges can be understood as the generalization issues within POMDPs. In this paper, our focus narrows to the generalization of a collection of POMDPs arising from an inherent MDP via diverse masked observation functions, which we refer to as a POMDP family.

Existing offline RL methods primarily target fully observable environments with comprehensive state data, causing difficulties in handling real-world scenarios involving partial observability and uncertain observation functions [30, 31, 29, 13, 28, 8, 15, 26, 7, 35]. While POMDP methods excel in online RL, they rely on continuous environment interactions for policy adaptation, making them unsuitable for offline RL [14, 47, 9, 21, 36, 20, 22, 32, 40]. Furthermore, most of these methods are developed for specific POMDPs and thus lack the ability to generalize across different and unknown partial observation functions. Although some attempts have been made to address partial observability in offline RL [17], their effectiveness is constrained by restrictive assumptions, such as a known observation function or a focus on theoretical analysis.

To bridge the research gap, we unveil Offline RL with DiscrEte pRoxy representation (ORDER), a framework that leverages discrete proxy state representations for diverse observation functions. Drawing an analogy, consider mapping a vast terrain: A detailed map is challenging to recall, but broadly categorizing into zones simplifies navigation. Similarly, the discrete representation in ORDER categorizes states, aiding RL algorithms. Moreover, even when part of the terrain (state) is obscured, a knowledgeable guide (proxy representation) can infer the underlying area. Our contributions are three-fold. Firstly, we propose a discrete representation of states to improve the robustness against the partial observability and a proxy representation to recover the discrete representation of the state from partial observable trajectory. Secondly, we propose a three-stage training of ORDER: i) we first train the discrete representation on data with full observation through Vector Quantized AutoEncoder (VQ-VAE [45]), ii) we train the decision module based on the discrete state, and iii) we train the proxy discrete representation on the data with various partial observations, aligning with the discrete representation. Finally, we present a thorough evaluation of ORDER, showcasing its effectiveness in enhancing offline RL algorithms for various partially observable scenarios. Our experiments highlight the importance of discrete proxy representations in maintaining alignment under partial observability, leading to improved generalization performance. ORDER is a flexible framework to employ any state-of-the-art (SoTA) offline RL algorithms and we hope ORDER can pave the way for the deployment of RL policy against various partial observabilities in the real world.

## 2 Related Works

**Offline RL.** Offline RL aims to learn policies from pre-collected transition data, eliminating the need for active data collection [34]. In recent years, this area has seen significant growth in research [30, 31, 29, 13, 28, 8]. Both empirical and theoretical studies have identified overestimation error, caused by out-of-distribution states and actions, as a critical challenge in offline RL [13]. As a result, numerous behavioral regularizations have been proposed to keep the learned policy within the bounds of the offline data during RL training [15, 26, 7, 35]. However, the majority of existing methods focus on fully observable environments, limiting their applicability in real-world situations that often involve partial observability and uncertain observation functions [31, 13, 28, 8, 26]. In such cases, the partial observability encountered during execution can greatly differ from the data on which the RL policy is trained, leading to a significant decline in performance. A noteworthy attempt to tackle partial observability in offline RL is the work by [17]. However, this effort mainly centers on theoretical analysis and assumes a known observation function, which restricts its practical application. Our work sets itself apart from this previous effort by assuming access to full observations in the offline dataset, while the observation functions in the testing environment are unknown, potentially diverse,

and subject to change. Our approach is designed to generalize to uncertain observation functions, thus improving generalization performance across a range of partially observable scenarios.

**Online RL for POMDPs.** Existing work addressing POMDPs generally falls into two main categories: i) belief state methods [27, 14, 47, 9, 21], which aim to deduce the distribution of hidden (belief) states from partial observation histories and then apply standard RL techniques to these inferred states; and ii) memory-based policies that use memory-based architectures, like recurrent neural networks (RNNs), to store observation trajectories and summarize past experiences [42, 3, 50, 24, 23, 37, 36, 49, 19]. However, most of these methods are developed for specific POMDPs and lack the ability to generalize across different and unknown partial observation functions. Furthermore, these online techniques hinge on the agent's capacity to interact with the environment and adjust its policy online, which is unsuitable for the offline RL setting. Another relevant area is asymmetric reinforcement learning, where existing works assume that agents can access privileged information (e.g., full observations) during training, but need to function in testing environments with partial observations [39, 1, 48, 2, 16, 33]. Nonetheless, they also assume a specific partial observation and depend on data collection through environment interaction. In contrast, our method presumes that we can access full observation information from a fixed dataset without interacting with the environment. Additionally, our method does not focus on a single, fixed partial observation function; it is designed to effectively generalize across a variety of masked observation functions. This adaptability allows our approach to address real-world situations characterized by diverse and uncertain observation functions.

**Sim-to-Real RL.** Sim-to-real RL is a notable direction in related works. This approach leverages simulation training for real-world applications, adapting policies based on transferred knowledge [52]. A prevalent strategy in this realm is domain randomization [44]. Rather than meticulously replicating real-world parameters, the environment is extensively randomized to encapsulate the genuine data distribution, regardless of simulation-reality bias. Additionally, many researchers adopt domain adaptation, which capitalizes on source domain data to enhance model performance in a target domain, typically data-scarce [6, 18, 25]. A key challenge here lies in unifying the often-divergent feature spaces between source and target domains to facilitate knowledge transfer effectively. Our work shares a thematic connection with the sim-to-real paradigm, as both address the transition from controlled conditions to real-world unpredictability. In our framework, agents, trained comprehensively on offline datasets, must adapt to partial real-world observations. While sim-to-real focuses on bridging the gap between simulation and reality, our emphasis is on transitioning from thorough training data to fragmented real-world observations. Both aim for real-world proficiency, but the challenges and remedies they explore differ substantially.

## 3 Preliminary

**POMDPs.** A Partially Observable Markov Decision Process (POMDP) is a mathematical framework used to model decision-making problems in uncertain environments [41]. Formally, a POMDP is defined as a tuple $(\mathcal{S}, \mathcal{A}, \mathcal{O}, T, S_0, O, O_0, R, H, \gamma)$, where the underlying process is a Markov Decision Process (MDP) $(\mathcal{S}, \mathcal{A}, T, S_0, R, H, \gamma)$. Concretely, $\mathcal{S}$ represents the set of states, $\mathcal{A}$ denotes the set of actions, $T : \mathcal{S} \times \mathcal{A} \times \mathcal{S} \to [0, 1]$ is the transition function (dynamics), $S_0 : \mathcal{S} \to [0, 1]$ specifies the initial state distribution, $R : \mathcal{S} \times \mathcal{A} \times \mathcal{S} \to \mathbb{R}$ is the reward function, $H \in \mathbb{N}$ refers to the time horizon, and $\gamma \in [0, 1)$ is the discount factor. Additionally, in the POMDP setting, $\mathcal{O}$ denotes the set of observations, and $O : \mathcal{S} \times \mathcal{O} \to [0, 1]$ represents the observation function. At the initial time step $t = 0$, a starting state $s_0 \sim S_0(\cdot)$ and an initial observation $o_0 \sim O_0(\cdot|s_0)$ are sampled. At any time step $t \in 0, \ldots, H - 1$, the policy takes an action $a_t \in \mathcal{A}$ in the environment, which updates the state according to the dynamics $s_{t+1} \sim T(\cdot|s_t, a_t)$. The next observation is then sampled as $o_{t+1} \sim O(\cdot|s_{t+1})$, and the reward is computed as $r_{t+1} = R(s_t, a_t, s_{t+1})$. Let the observable trajectory up to time step $t$ be denoted as $\tau_{0:t} = (o_0, a_0, o_1, \ldots, a_{t-1}, o_t, a_t)$, then the memory-based policy, in its most general form, is defined as $\pi(a_t|\tau_{0:t-1}, o_t)$, conditioned on the entire history.

In this paper, we focus on the *masked* partially observable setting, where state factors are obscured or "masked" in a dynamic manner, such as when sensors of a robot are occluded or signals are perturbed. In this context, any state can be represented as a collection of random variables $s_t = [s_t^1, s_t^2, \ldots, s_t^M]$. This implies that the state consists of various factors collected from multiple sources.

Each factor $s^i$ can differ in type and size, ranging from high-dimensional (e.g., multimedia data) to low-dimensional (e.g., tabular data). We introduce an $M$-dimensional binary mask variable $m \in \mathcal{M} = \{0, 1\}^{M \times H}$ to signify the observability of each factor: for the $t$-th time step and the $i$-th factor, $m_t^i = 1$ if it is observed, and $0$ otherwise. Consequently, we redefine $O$ as the *masked observation function*: Given the mask value $m$, the current observation $o_t$ can be get from $O(\cdot|s_t) = [s_t^i|m_t^i = 1]$. In this context, different mask variables result in distinct POMDPs due to variations in their observation functions. And the mask variable $m_t$ may change with the time step $t$. An MDP can be considered as a special case of a POMDP when the observation $o$ is always identical to the state $s$, i.e., for any time step $t$ and factor $i$, $m_t^i = 1$.

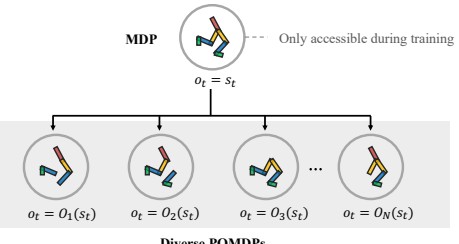

Figure 1: In general POMDP problem, the policy is trained on samples which contain state information and expected to perform well across various POMDPs with distinct observation functions.

**Generalization over a POMDP Family.** Prior works generally aim to address a specific POMDP, where the policy is trained and tested within the same POMDP. In this paper, we aim to tackle a more general POMDP problem as depicted in Figure 1. In particular, during the training period, the policy has access to samples which include state information. During the testing period, the policy is expected to perform well in various POMDPs that share the same underlying MDP but have distinct observation functions. The objective for such a problem in the masked POMDP setting can be regarded as finding an optimal policy $\pi(a_t|\tau_{0:t})$ to maximize the expected discounted return under different masked observation functions: $\max_\pi \mathbb{E}_{s_0 \sim S_0, m \sim \mathcal{M}}[\mathbb{E}_\tau[\sum_{t=0}^{H-1} \gamma^t r_{t+1}|s_0]]$.

**Offline RL.** Our work focuses on the offline setting, where the agent learns a policy from a static dataset $\mathcal{D} = (s, a, s', r)$ without collecting new experience data. The dataset is generated by another policy, referred to as the *behavioral* policy and denoted by $\pi_\beta(a|s)$. We allow full access to state information during the training phase by relaxing partial observability constraints. However, during execution, the trained policy must handle diverse partial observation functions. Our method separates representation learning and policy learning, enabling the use of state-of-the-art offline RL algorithms to train a decision head that can make decisions based on the learned representation (see Section 4). In our paper, we employ Implicit Q-learning (IQL) [28] (see details of IQL in Appendix.) as the offline RL algorithm for training our decision module.

## 4 Offline RL with Discrete Proxy Representations

We introduce ORDER, a framework designed to tackle challenges in partially observable settings for offline RL. Our approach uses two main strategies: i) We learn discrete proxy state representations to handle out-of-distribution states and uncertainties in observation functions. The choice of discrete state representation is deliberate, as it ensures that our policy maintains its generalizability even in situations characterized by uncertainty. By converting unseen partial observations into discrete forms akin to those present in the training data, ORDER could adapt to a variety of uncertain, partially observable situations. ii) We use an estimated proxy representation to connect partial observations with learned discrete state representations. This allows our model to understand the state information even with limited observations, improving accuracy and making the optimization process easier. In the following subsections, we first give an overview of the training process, then explain how we learn discrete state representations and discrete proxy representations for partial observations.

**Overview.** As illustrated in Figure 2, ORDER is a three-phase paradigm for developing a policy capable of addressing the general POMDP problem, In the first phase, we train a state encoder $\Phi$ for generating *discrete* state representations and then convert the original dataset $\mathcal{D} = \{(s, a, s', r)\}$ to a new dataset $\mathcal{D}_z = \{(z, a, z', r)\}$, where $z = \Phi(s) \in \mathcal{Z}$ is the concatenation of representations from different state factors. We will introduce the details of $\Phi$ in the following subsection. In the second phase, an arbitrary offline RL algorithm is applied to learn an **decision head** $h(a|z)$ from $\mathcal{D}_z$. This head could provide an action for a given state representation $z$. Combined with $\Phi$, the **oracle policy** $\pi(a|s)$, which can be executed on the state space, can be written as $\pi(a|s) = h\left(a|\Phi(s)\right)$. However,

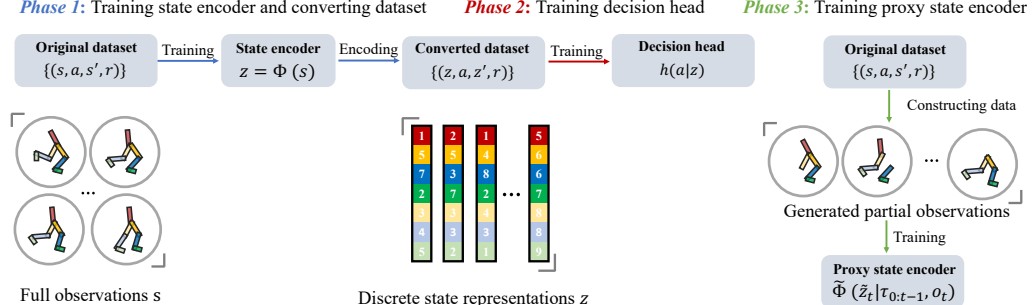

Figure 2: The three-phase training paradigm of ORDER, encompassing **(1)** training a state encoder $\Phi$, **(2)** training a decision head $h(a|z)$, and **(3)** training a proxy state encoder $\tilde{\Phi}$.

the full state information is only available during training. During execution, the overall policy cannot be directly used, since some state factors will be missed. To address this issue, in the third phase, we train a *proxy* state encoder $\tilde{\Phi}$ to predict the discrete state representation $z$ from the past history of partial observations instead of the full state information: $\tilde{z}_t \sim \tilde{\Phi}(\cdot|\tau_{0:t-1}, o_t)$ . Here, $\tilde{z}_t$ is the predicted state representation which is conditioned on the past history of observations. Combining the decision head $h(a|z)$ with the proxy state encoder $\tilde{\Phi}(\cdot|\tau_{0:t-1}, o_t)$, we obtain a **proxy policy**, which could address different partially observable scenarios, as $\tilde{\pi}(a_t|\tau_{0:t-1}, o_t) = h\left(a|\tilde{\Phi}(\cdot|\tau_{0:t-1}, o_t))\right)$. We summarize the oracle policy and proxy policy in Figure 3 (c-d).

## 4.1 Learning Discrete Representations over States

In this subsection, we present a method for learning discrete state representations, drawing inspiration from the VQ-VAE approach [45]. Recall that in our setting, we consider the state as a collection of $M$ heterogeneous state factors: $s = [s^1, s^2, \cdots, s^M]$ (refer to Section 3). For each factor, we define a codebook $e^i \in \mathbb{R}^{K \times D}, i \in \{1, 2, \cdots, M\}$, where $K$ represents the size of the discrete latent space (i.e., a $K$-way categorical), and $D$ denotes the dimensionality of each latent embedding vector $e_j^i \in \mathbb{R}^D, j \in \{1, 2, \cdots, K\}$. As shown in Figure 3 (a), the factor encoder $\phi^i$ takes the state factor $s^i$ as input and outputs an embedding $\hat{e}^i$. The discrete latent variable $z^i$ is then calculated by the nearest neighbor look-up $g$ using the codebook $e^i$ as follows:

$$z^i = g(\hat{e}^i, e^i) = e_j^i \qquad \text{where } j = \arg\min_j \|\hat{e}^i - e_j^i\|_2. \qquad (1)$$

Next, we obtain the discrete compositional representation $z$ by concatenating the discrete representations of all state factors:

$$z = \Phi(s) = \text{CONCAT}(z^1, z^2, \cdots, z^M). \qquad (2)$$

Here, we summarize the entire process of mapping $s$ to $z$ as $z = \Phi(s)$. For simplicity, we refer to $\Phi$ as the *state encoder*, which actually comprises the set of all codebooks and factor encoders: $\Phi = \{e^i, \phi^i | i \in 1, 2, \cdots, M\}$. And its architecture is illustrated in Figure 3 (a).

**Learning Objective for $\Phi$.** The representation learning objective is based on the Vector Quantized AutoEncoder (VQ-VAE) approach [45] and can be expressed as follows:

$$\mathcal{L}(\Phi, \Sigma) = \mathbb{E}_{s \sim \mathcal{D}} \left( \left|\Sigma(z) - s\right|_2^2 + \frac{1}{M} \sum_i^M \left|\text{sg}(\hat{e}^i) - z^i\right|_2^2 + \frac{\beta}{M} \sum_i^M \left|\hat{e}^i - \text{sg}(z^i)\right|_2^2 \right). \qquad (3)$$

Here, $\Phi$ represents the set of all codebooks and factor encoders, while $\Sigma$ denotes the state decoder and $\Sigma(z)$ is the state reconstruction. The function $\text{sg}(\cdot)$ performs a stop-gradient operation to block gradients from entering its argument. The hyperparameter $\beta$ controls the degree to which the code can change. The first term is the reconstruction loss, which ensures that the encoder captures sufficient state information. Note that for the nearest lookup operation in Equation (1), there is no actual gradient, so we use the straight-through estimator [4] to copy gradients from the decoder input $z^i$

to the encoder output $\hat{z}^i$. The second term is the codebook loss, which only applies to the discrete latent vector and encourages the chosen $z^i$ to be close to the encoder output $\hat{z}^i$. The third term is the commitment loss, which only applies to the factor encoders and encourages $\hat{z}^i$ to remain close to the selected discrete latent vector $z^i$. We set the value of $\beta$ to 0.25, as in the original VQ-VAE paper [45]. The training paradigm of state encoder is summarized in Appendix.

## 4.2 Learning Discrete Proxy Representations over Partial Observations

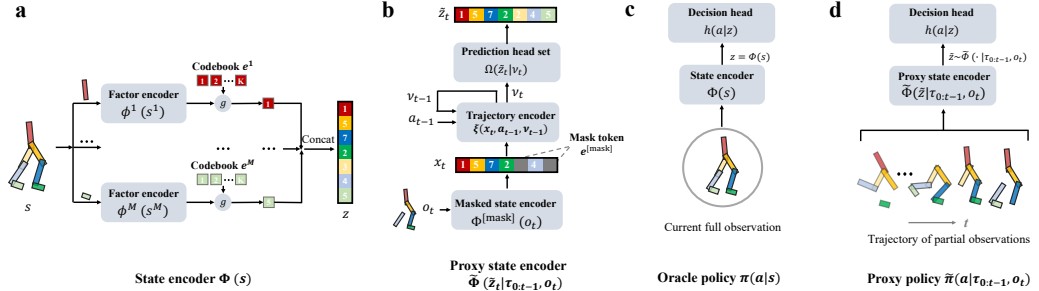

Figure 3: The architectures of (**a**) the state encoder $\Phi(s)$ and (**b**) the proxy state encoder $\tilde{\Phi}(\tilde{z}|\tau_{0:t-1}, o_t)$. And the constructions of (**c**) the oracle policy $\pi(a|s)$ and (**d**) the proxy policy $\tilde{\pi}(a_t|\tau_{0:t-1}, a_t)$.

In this section, we focus on the alignment of partial observations with the learned discrete state representations derived from full state information. To achieve this, we propose a method that employs a recurrent architecture and a learnable mask token, allowing the model to handle partial observability and estimate the underlying state of the environment. By aligning the estimation process with the learned representations mentioned in the previous subsection, our model can effectively generate proxy state representations from incomplete observations, enabling efficient decision-making in partially observable settings. Overall, our aim is to obtain an inference model named *proxy state encoder*, denoted as $\tilde{\Phi}(\tilde{z}|\tau_{0:t-1}, o_t)$, which approximates the true posterior distribution $p(z_t|\tau_{0:t-1}, o_t)$. Its architecture is illustrated in Figure 3 (b).

### 4.2.1 Construction of Partial Observable Training Data

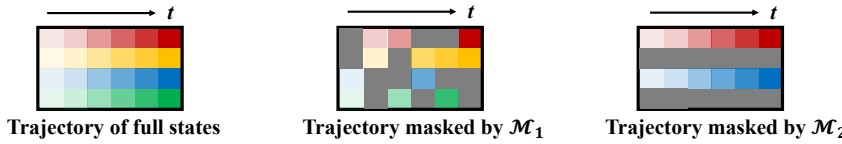

Figure 4: **Illustration of Both Missing Scenarios.** The figure showcases trajectories spanning various time steps (horizontal axis) and state factors (vertical axis). Colorful grids represent observable state factors, while grey grids signify masked state factors. The **left** panel presents the trajectory of the full states. In contrast, the **middle** and **right** panels display trajectories following masking by the dynamical missing scenario $\mathcal{M}_1$ and the factor reduction scenario $\mathcal{M}_2$, respectively.

The first step to training $\tilde{\Phi}$ is to get a good training dataset. However, one challenge is that we cannot directly collect partial observations from the environment in the offline RL setting. We only have the original offline dataset $\mathcal{D} = \{(s, a, s', r)\}$ and the converted representaion dataset $\mathcal{D}_z = \{(z, a, z', r)\}$ that contains full state information. To address this issue, we construct trajectories of partial observations by utilizing the converted dataset as the training data for $\tilde{\Phi}$. And we introduce partial observability by randomly setting a portion of mask variables $m_t^i$ to be 0, simulating real-world partially observable scenarios. Specifically, we adopt two scenarios :

- **Dynamical Missing Scenario $\mathcal{M}_1$:** For each time step $t$, we set the mask variable $m_t^i = 0$ at the probability of $\eta$, and otherwise at the probability of $1 - \eta$. In this case, the missing factors dynamically change over time with a certain missing ratio.

- **Factor Reduction Scenario $\mathcal{M}_2$ :** Given a trajectory of data whose time length is $T$, we set the mask variables $m_{0:T}^i = 0$ at the probability of $\eta$, and otherwise at the probability of $1 - \eta$. This scenario arises when part of the sensors are removed, and the corresponding components of the observations are consistently missing over time.

We give examples of trajectories masked by both missing scenarios in Figure 4. And we adopted both scenarios in our training paradigm by using them iteratively, and denoted the process generating random masking variables as $m_{0:t} \sim \mathcal{M}_\eta$, which is summarized in Appendix.

### 4.2.2 Learning Discrete Proxy Representations

After generating training data, the next step to training $\tilde{\Phi}$ is to aggregate information in partial observable trajectory. Concretely, given the partial observation $o = [s^i | m^i = 1]$,

$$x = \Phi^{[\texttt{mask}]}(o) = \text{CONCAT}(x^1, x^2, \cdots, x^M) \text{ where } x^i = \begin{cases} z^i = g^i\Big(\phi^i(s^i)\Big) & \text{if } m^i = 1 \,, \\ e^{[\texttt{mask}]} & \text{if } m^i = 0 \,. \end{cases} \quad (4)$$

Here, we denote $\Phi^{[\texttt{mask}]}$ the masked state encoder, which contains the trained state encoder mentioned in the previous subsection and an additional learnable mask token $e^{[\texttt{mask}]} \in \mathbb{R}^{1 \times D}$. It directly converts the observable state factor into the learned discrete factor representation, and represents the unobservable state factor as the mask token. Next, we use a trajectory encoder $\xi$ to capture the temporal information in the trajectory of partial observations. Concretely, it takes current observation representations and previous actions as input and output a continuous vector $\nu_t = \xi(x_t, a_{t-1}, \nu_{t-1})$.

**Discrete State Representation Distribution Estimation.** Now, we want to predict the categorical distribution of $p(z_t | \tau_{0:t-1}, o_t)$ . For each factor $i$, we build a set of prediction head $\Omega = \{\omega^i | i = 1, 2, \cdots, M\}$ which takes the trajectory representation $\nu_t$ as input, and output the categorical probability over the discrete embeddings of the unobserved factors in the form of softmax output. And we only estimate the unobservable factor.

During the execution, we predict the proxy state representations as follow:

$$\widetilde{z} = \text{CONCAT}(\widetilde{z}^1, \widetilde{z}^2, \cdots, \widetilde{z}^M) \text{ where } \begin{cases} \tilde{z}^i = z^i & \text{if } m^i = 1 \\ \tilde{z}^i \sim \omega^i(\cdot | \nu_t) & \text{if } m^i = 0 \end{cases} \quad (5)$$

**Learning Objective for $\tilde{\Phi}$.** The overall learning objective for the distribution estimation can be summarized as minimizing the Cross-Entropy loss between the predicted probability distribution and the true discrete embedding associated with the input sample:

$$\mathcal{L}(e^{[\texttt{mask}]}, \xi, \Omega) = - \mathop{\mathbb{E}}_{t \sim U(0, H-1), z_{0:t}, a_{0:t-1} \sim \mathcal{D}_z, m_{0:t} \sim \mathcal{M}_\eta} \Big[ \frac{1}{M} \sum_i^M (1 - m_t^i) \log \omega^i(\tilde{z}^i = z^i | \nu_t) \Big]. \quad (6)$$

The training paradigm of proxy state encoder is summarized in Appendix.

## 5 Experiments

In this section, we present a thorough empirical evaluation of ORDER, highlighting its effectiveness in improving offline RL algorithms for a variety of partially observable scenarios. We integrate ORDER with the IQL algorithm [28] and assess its performance across an array of environments in the D4RL benchmark [12] under different partial observation situations. Our results clearly indicate that ORDER substantially enhances the generalization performance of policies trained on offline datasets in diverse partially observable conditions. To ensure the robustness of our findings, we conduct all experiments with five distinct random seeds, each consisting of ten separate runs. For detailed information on the architectures and hyperparameters, please refer to Appendix. In the following subsections, we first evaluate the generalization performance of our model and other baselines under varying degrees of partial observability. Subsequently, we carry out an empirical study to analyze the significance of our learned discrete proxy representations.

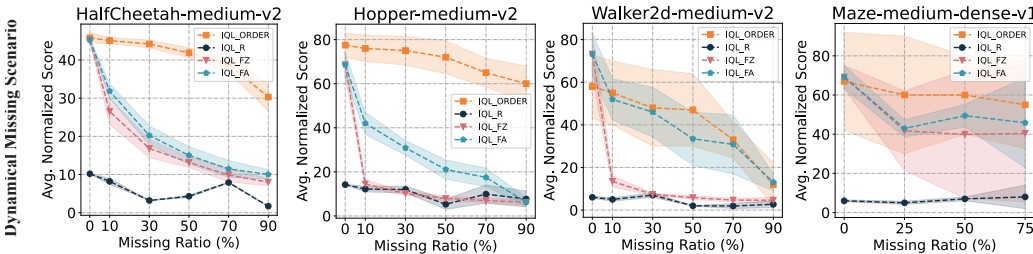

Figure 5: **Generalization performance of different models under dynamic missing scenarios.**

## 5.1 Evaluating Generalization Performance under Diverse Partial Observation Functions

In this subsection, our primary objective is to assess the generalization performance of policies, which are trained on offline datasets containing full observations, under various partial observation scenarios. We evaluate ORDER and other baseline algorithms on gym locomotion tasks and maze navigation tasks in the D4RL benchmark [12]. In our experimental setup, we treat each dimension of the observation space as an independent state factor.

Our proposed model employs the IQL algorithm [28] to train the decision head on the representation space. By combining the trained proxy state encoder with the decision head, our model can make decisions under partial observations. We denote it as **IQL_ORDER** and compare it with the following baselines: i) IQL with the recurrent architecture (**IQL_R**): We integrate IQL with a state-of-the-art method for online RL in POMDPs [40] to create this baseline. This method utilizes a specialized recurrent neural network architecture to tackle partial observability and can be combined with arbitrary actor-critic algorithms. During training, we employ the same mask strategy (see Section 4.2.1) to construct partial observable training data for it. Furthermore, to indicate which state factors are missing, we use 0 to replace with the value of unobserved factors and concatenate the mask variable $m_t$ with the observation vector $o_t$ as the extended input. ii) IQL with the strategy of filling zero (**IQL_FZ**): This baseline is trained directly using the IQL algorithm with the original offline dataset. During testing, 0 are used to represent unobservable factors. iii) IQL with the strategy of filling adjacent elements (**IQL_FA**): This baseline is also trained directly using the IQL algorithm with the original offline dataset. During testing, if there is a latest observed value of an unobserved factor, it is used to represent the current unobservable factors; otherwise, 0 is used.

**Generalization Performance under Dynamical Missing Scenarios.** We first evaluate policy performance under dynamically missing scenarios, where certain state factors may intermittently and randomly be absent over time. We conduct this evaluation across various missing ratios to gauge the models' adaptability. An intrinsic characteristic of a policy with strong generalization ability is its ability to maintain performance as the missing ratio increases. The results are illustrated in Figure 5. IQL_ORDER outperforms other baselines in most cases, demonstrating its superior generalization performance over dynamic missing scenarios. On the other hand, the performance of IQL_ORDER slightly decreases with increasing missing ratios in most cases. However, even in extreme missing scenarios (missing ratio greater than 70%), it still maintains good performance. Additionally, IQL_FZ and IQL_FA exhibit similar performance, with IQL_FA slightly outperforming IQL_FZ. Their performance is worse than IQL_ORDER, indicating that simply padding zeros or the latest values into unobservable values cannot effectively address this task. Filling adjacent elements proves to be a better strategy than filling zeros. Finally, IQL_R performs the worst in all cases under dynamic missing scenarios. However, it is worth noting that IQL_R performs well in settings where observation functions are single and fixed (see Appendix). This suggests that it struggles to develop effective policies under diverse and dynamic partial observation settings. One possible reason is that the diverse partial observation function leads to enhanced non-stationarity of dynamics, making it challenging to directly train policies on these cases and resulting in highly unstable training.

**Generalization Performance under Factor Reduction Scenarios.** Next, we evaluate policy performance under factor reduction scenarios, where in each test episode, some state factors are randomly removed throughout the entire episode. Notably, we exclude the IQL_FA model in this task since we cannot obtain the latest values of the missing factors.

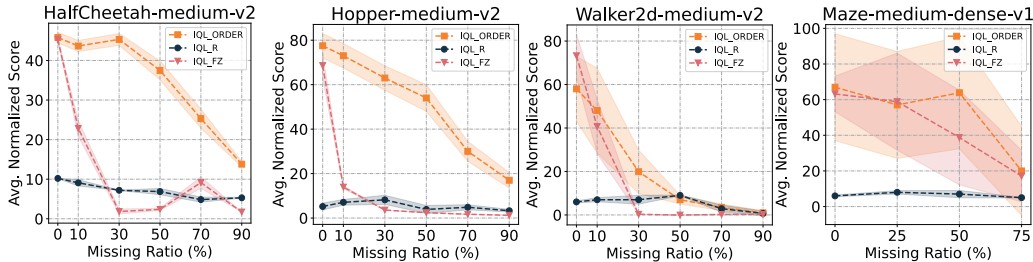

Figure 6: **Generalization performance of different models under factor reduction scenarios.**

The results are depicted in of Figure 6. IQL_ORDER outperforms other baselines in most cases. However, when the missing ratio is high, its performance decreases more significantly than in dynamic changing scenarios. This might be attributed to the increased difficulty of factor reduction scenarios compared to dynamic changing scenarios. In this setting, models cannot infer the current missing factors from their latest observed values; instead, they must rely on other observable factors for inference. Both IQL_FZ and IQL_R continue to perform poorly, indicating that padding zeros is not an effective strategy and that IQL_R cannot achieve stable training under such challenging scenarios.

## 5.2 Empirical Analysis of Discrete Proxy Representations

In this subsection, we aim to underscore the significance of discrete proxy representations in our framework. We conduct experiments on the `Hopper-medium-v2` dataset. For these experiments, we set up two comparison models—**Continuous Reps.** and **Raw States**—both of which employ our three-phase paradigm to develop policies. The **Continuous Reps.** model uses a Variational Autoencoder (VAE) in the first phase to generate continuous representations, and in the third phase, it predicts Gaussian distributions over these continuous representations. On the other hand, the **Raw States** model applies an identity function as its state encoder in the first phase and predicts Gaussian distributions of the missing factors in the third phase.

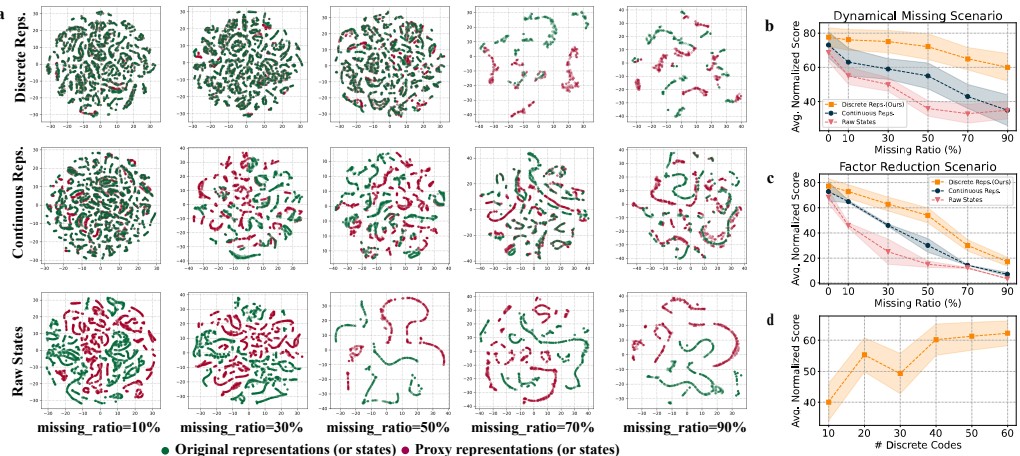

Figure 7: **Empirical Analysis of discrete proxy representations.** (**a**) T-SNE visualization showing alignment of discrete, continuous, and raw state representations with original representations under varying missing ratios in Factor Reduction Scenarios. (**b-c**) Generalization performance under dynamic missing and factor reduction scenarios. (**d**) Performance evaluation with varying numbers of learned discrete representations.

One challenge we would like to address is the potential misalignment between the partial observability encountered during testing and the data on which the RL policy is trained. To examine this, we first sample 10 trajectories generated by different models under various missing ratios in the Factor Reduction Scenarios. We then use t-SNE [46] to visualize their generated proxy representations (or states) and the original representations (or states), which can be regarded as the ground truth under full

observability. The proxy representations are intended to prevent policies from encountering unseen observations, which could lead to severe performance degradation in the offline RL setting. Therefore, a suitable proxy representation should align well with the original representation, even when the missing ratio is high. As shown in Figure 7 (a), we observe that discrete proxy representations align well with the original representations when the missing ratio is below $70\%$. In contrast, continuous proxy representations begin to mismatch with the original representations when the missing ratio exceeds $10\%$. Meanwhile, the proxy raw states show a mismatch at a missing ratio of $10\%$. Furthermore, Figure 7 (b-c) illustrates the generalization performance under both missing scenarios. We find a positive correlation between alignment ability and generalization performance: discrete representations outperform continuous representations, which in turn perform better than raw states. Additionally, discrete representations maintain good performance when the missing ratio is below $70\%$, but decrease when the missing ratio exceeds $70\%$ in the Factor Reduction Scenario, consistent with the results in Figure 7 (a). These findings emphasize the importance of utilizing discrete proxy representations in our framework for maintaining a strong alignment between proxy and original representations under partial observability. This alignment allows the policy to make better decisions and exhibit improved generalization performance in partially observable settings. We also evaluate the performance of models with different numbers of learned discrete representations, as shown in Figure 7 (d). We find that increasing the number of discrete representations can improve model performance, likely because a higher number of discrete codes enhance the model's expressive ability. However, when the number exceeds $40$, the performance improvement is marginal, indicating that a trade-off between the number of discrete codes and model performance can be achieved.

## 6 Conclusion

We present ORDER, a novel probabilistic offline RL framework that addresses the challenges of partial observability in real-world scenarios. By leveraging discrete proxy state representations, ORDER significantly improves robustness and generalization performance across diverse observation functions. Our three-stage training approach effectively harnesses the power of discrete state representations and proxy discrete representation alignment. Extensive experiments demonstrate the effectiveness of ORDER in various partially observable settings. However, some limitations still exist. ORDER currently focuses on masked partial observation functions, leaving other forms like perturbations and noises unaddressed. Moreover, its applicability to more complex real-world tasks, such as autonomous driving, remains to be explored. Despite these limitations, we hope ORDER can pave the way for the deployment of RL policy against various partial observabilities in the real world.

## 7 Acknowledgements

This research is supported by the National Research Foundation, Singapore under its Industry Alignment Fund – Pre-positioning (IAF-PP) Funding Initiative. Any opinions, findings and conclusions or recommendations expressed in this material are those of the author(s) and do not reflect the views of National Research Foundation, Singapore.

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
