# A   Architectures, Hyper-parameters and Algorithms

Our approach, named ORDER, uses a three-step training process. It involves: **(1)** Training a state encoder $\Phi$; **(2)** Training a decision head $h(a|z)$; and **(3)** Training a proxy state encoder $\tilde{\Phi}$.

In the next parts of this section, we'll explain the methods, structures, and settings we use in each of these steps. After that, we'll talk about how we set up and carried out our experiments. This section is set up to help readers understand the three steps of ORDER, and how we put it into practice in our experiments.

## A.1   State Encoder

In this section, we'll break down the design of the state encoder, how we decided on the best hyper-parameters, and the process we used to train the state encoder.

We used a grid search strategy to find the optimal hyper-parameters for our experiments. The selection of these parameters can be found in Table 1.

Table 1: Choosing hyper-parameters for training the state encoder.

| Hyper-parameter | Final choice |
| --- | --- |
| Codebook dimension | 2 |
| # of discrete codes | 40 |
| Feedforward dim of $\Sigma$ | 256 |
| Training steps | $2 \times 10^5$ |
| Learning rate | $5 \times 10^{-3}$ |
| Batch size | 1024 |

The state encoder $\Phi$ in ORDER consists of a set of codebooks and factor encoders: $\Phi = \{e^i, \phi^i | i \in 1, 2, \cdots, M\}$. For our experiments, we made $M$ the same as the number of dimensions in the observation. This allowed each observation dimension to match up with a state factor. We represented each factor encoder with a fully-connected layer, which means it has an input dimension of 1 and an output dimension equal to the codebook dimension.

During the training of the state encoder, we used a state decoder $\Sigma(z)$ to calculate the reconstruction term in the loss function (Equation (3) in the main text). The state decoder architecture is a three-layer MLP. Its input dimension equals the state representation dimension (which is the number of state factors times the dimension of the discrete code), and its output dimension is equal to the state dimension. We tried values of $[64, 128, 256, 512]$ for the feedforward dimension, and ultimately chose 256. For the commitment loss term $\beta$, we set it to $0.25$, keeping it consistent with the original VQVAE paper [4]. We summarize the training process in Algorithm 1.

## A.2   Decision Head

In the second phase of training, ORDER employs an arbitrary offline RL algorithm to instruct a decision head $h(a|z)$. The decision head is tasked with making decisions based on provided discrete state representations. These state representations are derived from a transformed dataset, denoted as $\mathcal{D}_z = \{z, a, z', r\}$. Here, $z$ represents the discrete state representation generated by $z = \Phi(s)$. Thus, the decision head is guided by these state representations to execute actions effectively.

ORDER has the flexibility to improve policy performance when added to existing offline RL algorithms, especially in situations dealing with different partially observable functions. In the second phase of our approach, we trained the decision head $h(a|z)$ using Implicit Q learning (IQL) [2], an established algorithm that we selected for this study. We're optimistic that by combining our method with other cutting-edge offline RL algorithms, we could further elevate the results. We aim to investigate these possibilities in future research. In this section, we'll discuss the IQL algorithm and elaborate on the specific architecture and hyper-parameters we employed in our project.

IQL is a method used in offline reinforcement learning. It tackles two main goals: improving the policy that guides decision-making and limiting changes from the original policy to avoid mistakes

---
**Algorithm 1:** Training state encoder $\Phi$
---
**Data:** Offline dataset $\mathcal{D}$.
**Result:** State Encoder $\Phi = \{e^i, \phi^i | i \in 1, 2, \cdots, M\}$.
$N \leftarrow n$;     # The number of trainning steps
Initialize the state encoder $\Phi$;
Initialize the state decoder $\Sigma$;
**while** $N \neq 0$ **do**
    Sample states: $s \sim \mathcal{D}$;
    **for** $i = 1, 2, \cdots, M$ **do**
        compute factor embeddings for each state factor: $\hat{e}^i = \phi^i(s^i)$;
        get the discrete latent variable for each factor by nearest lookup: $z^i = g(\hat{e}^i, e^i)$;
    **end**
    Construct the discrete state representation: $z = \mathrm{CONCAT}(z^1, z^2, \cdots, z^M)$;
    Compute the reconstruction by state decoder: $\Sigma(z)$;
    Compute the loss according to Equation (3) in the main text;
    Backpropagate gradients to all parameters for minimizing the loss;
    $N \leftarrow N - 1$;
**end**
---

due to shifts in the data distribution. The key idea in IQL is to view the value of a state (a measure of how good it is to be in that state) as something that can vary based on the action taken. By doing this, IQL can improve the policy without ever needing to consider actions that are not in the original dataset.

The method works by switching between fitting a value function (which estimates the value of the best possible actions for a state):

$$L_V(\psi) = \mathbb{E}_{(z,a) \sim \mathcal{D}_z}[L_2^\tau Q_{\hat{\theta}}(z,a) - V_\psi(z)]. \tag{1}$$

and translating it into a Q-function, which doesn't require a direct policy:

$$L_Q(\theta) = \mathbb{E}_{(z,a,z',r) \sim \mathcal{D}_z}[r + +\gamma V_\psi(z') - Q_{\hat{\theta}}(z,a)]^2. \tag{2}$$

The decision head is then created from this Q-function, without needing to consider actions not in the original dataset.

$$L_h(\delta) = \mathbb{E}_{(z,a) \sim \mathcal{D}_z}\Big[\exp\Big(\beta(Q_{\hat{\theta}}(z,a) - V_\psi(z))\log h_\delta(a|z)\Big)\Big]. \tag{3}$$

IQL is easy to use and efficient, needing only an additional part that fits with an asymmetric L2 loss (a type of error measurement). We summarize IQL in Algorithm 2 and present the choice of hyper-parameters in Table 2.

---
**Algorithm 2:** Training decision head by IQL [2]
---
**Data:** Converted offline dataset $\mathcal{D}_z$.
**Result:** Decion head $h_\delta(a|z)$
$N \leftarrow n$;     # The number of trainning steps
Initialize parameters $\psi, \theta, \hat{\theta}, \delta$;
**while** $N \neq 0$ **do**
    Train value function: $\psi \leftarrow \lambda_V \nabla L_V(\psi)$;
    Train Q function: $\theta \leftarrow \lambda_Q \nabla L_Q(\theta)$;
    $\hat{\theta} \leftarrow (1 - \alpha)\hat{\theta} + \alpha\theta$;
    Train decision head: $\delta \leftarrow \lambda_h \nabla L_h(\delta)$;
    $N \leftarrow N - 1$;
**end**
---

Table 2: Choosing hyper-parameters for training the decision head.

| Hyper-parameter | Final choice |
| --- | --- |
| # hidden layers in $Q_{\hat{\theta}}$, $V_\psi$ and $h_\delta$ networks | 2 |
| Dimension of hidden layers | 256 |
| Feedforward dimension of $\Sigma$ | 256 |
| Training steps | $1 \times 10^6$ |
| Sampled context length | 64 |
| Batch size | 64 |
| $\alpha$ | 0.05 |
| $\tau$ | 0.7 |
| $\beta$ | 3.0 |
| $\lambda_Q, \lambda_V, \lambda_h$ | $3 \times 10^{-4}$ |

## A.3 Proxy State Encoder

In this section, we first explain the procedure for generating random masking variables, followed by an outline of the architecture, hyper-parameters, and algorithms for the proxy state encoder.

Initially, the process of generating masking variables is denoted as $m_{0:t} \sim \mathcal{M}_\eta$ and summarized in Algorithm 3. This notation provides a concise mathematical description of the masking variable generation process, aiding in understanding the procedure involved.

---

**Algorithm 3:** Generating random mask variables

---

**Data:** trajectory length $t + 1$, factor missing ratio $\eta$, the number of state factos $M$.
**Result:** mask variables $m_{0:t}$
initialize mask variables as a zero matrix: $m_{0:t} = \mathbf{0}_{M \times (t+1)}$ ;
$p \sim U(0,1)$; # Sample a random variable to determine which scenario would be adopted
**if** $p \leq 0.5$ **then**
    # adopt the dynamical missing scenario
    **for** $i = 1, 2, \cdots, M$ **do**
        **for** $j = 0, 1, \cdots, t$ **do**
            $q \sim U(0,1)$;
            **if** $q \leq \eta$ **then**
                $m_j^i = 1$
            **end**
        **end**
    **end**
**else**
    # adopt the factor reduction scenario
    **for** $i = 1, 2, \cdots, M$ **do**
        $q \sim U(0,1)$;
        **if** $q \leq \eta$ **then**
            $m_{0:t}^i = \mathbf{1}_{1 \times (t+1)}$
        **end**
    **end**
**end**

---

Now, we delve into the architecture of the proxy state encoder. We start by initializing a random embedding to serve as the learnable mask token. This token has the same dimension as the discrete factor code. We then use a Gated Recurrent Unit (GRU) network [1] with a hidden layer as the trajectory encoder, denoted as $\xi$. This encoder takes the current observation representation and the action from the previous time step as inputs and generates a trajectory representation as its output. Following this, the prediction set, symbolized as $\Omega = \{\omega^i | i = 1, 2, \cdots, M\}$, comprises a series of linear

layers. These layers take the trajectory representation as input and produce a categorical distribution over the discrete codes. This process is accomplished by injecting their output into a softmax function. The procedure for training the proxy state encoder is outlined in Algorithm 4. We also detail our final selection of hyper-parameters in Table 3. This systematic approach aids in achieving an effective training process for the proxy state encoder.

Table 3: Choosing hyper-parameters for training the proxy state encoder.

| Hyper-parameter | Final choice |
|---|---|
| Factor missing ratio $\eta$ | 0.5 |
| Dimension of mask token | 2 |
| Dimension of hidden layer in $\xi$ | 128 |
| Training steps | $2 \times 10^5$ |
| Batch size | 64 |
| Learning rate | $1 \times 10^{-3}$ |

---

**Algorithm 4:** Training proxy state encoder $\tilde{\Phi}$

---

**Data:** Offline dataset $\mathcal{D}$, state encoder $\Phi$, horizon length $H$, factor missing ration $\eta$, the number of state factors $M$.
**Result:** State Encoder $\tilde{\Phi} = \{e^{[\text{mask}]}, \Phi, \xi, \Omega\}$.
$N \leftarrow n$;      # The number of trainning steps
Initialize the learnable mask token $e^{[\text{mask}]}$;
Initialize the trajectory encoder $\xi$;
Initialize the prediction head set $\Omega$;
**while** $N \neq 0$ **do**
    Sample the trajectory length: $t \sim U(0, H-1)$;
    Sample the random mask variable: $m_{0:t} \sim \mathcal{M}_\eta$;
    Sample the trajectory: $\tau_{0:t} \sim \mathcal{D}$
    Sample states: $s_{0:t} \sim \mathcal{D}$;
    initialize the action as a zero vector: $a_{-1} = \mathbf{0}$;
    initialize the trajectory representation as a zero vector: $\nu_{-1} = \mathbf{0}$;
    **for** $n = 0, 1, \cdots, t$ **do**
        Compute the partial observation representation according to Equation (4) in the main text:
        $x = \Phi^{[\text{mask}]}(s_n, m_n)$;
        Compute the trajectory representation: $\nu_n = \xi(x_n, a_{n-1}, \nu_{n-1})$;
        **for** $i = 1, 2, \cdots, M$ **do**
            Compute the true discrete code of factor $i$: $z^i = g^i\left(\phi^i(s^i), e^i\right)$ ;
            Infer the discrete code of factor $i$: $\tilde{z}^i \sim \omega^i(\cdot|\nu_n)$;
        **end**
    **end**
    Compute the loss: $\left[\frac{1}{M}\sum_i^M (1 - m_t^i)\log \omega^i(\tilde{z}^i = z^i|\nu_t)\right]$;
    Backpropagate gradients to all parameters for minimizing the loss;
    $N \leftarrow N - 1$;
**end**

---

# B   Additional Experimental Settings

In this section, we provide more details about our experimental setup and share some extra experimental outcomes. To make sure our results are reliable, we run all experiments with 5 different random seeds, and each of these seeds is used in 10 individual runs. Also, for every test episode, we set the maximum number of steps per episode to 1000.

Table 4: Choosing hyper-parameters for training the IQL_R.

| Hyper-parameter | Final choice |
|---|---|
| # hidden layers in $Q$, value, and policy networks | 2 |
| Dimension of hidden layers in GRU networks | 128 |
| Dimension of DQN networks | $[256, 256]$ |
| Dimension of value networks | $[256, 256]$ |
| Dimension of policy networks | $[256, 256]$ |
| Action embedding size | 16 |
| Observation embedding size | 32 |
| Reward embedding size | 16 |
| Training steps | $1 \times 10^6$ |
| Sampled context length | 64 |
| Batch size | 64 |
| $\alpha$ | 0.05 |
| $\tau$ | 0.7 |
| $\beta$ | 3.0 |
| $\lambda_Q, \lambda_V, \lambda_h$ | $3 \times 10^{-4}$ |

Table 5: Average normalized score of our model and IQL_R under single specific observation functions.

| Observation function | Dataset | IQL_ORDER | IQL_R |
|---|---|---|---|
| Mask-P | HalfCheetah-medium-v2 | 35.35 | 33.76 |
| | Hopper-medium-v2 | 75.23 | 70.23 |
| | Walker2d-medium-v2 | 7.35 | 3.45 |
| Mask-V | HalfCheetah-medium-v2 | 42.10 | 40.12 |
| | Hopper-medium-v2 | 54.41 | 44.56 |
| | Walker2d-medium-v2 | 7.34 | 14.23 |

**Sampling Various Partial Observation Functions.** In our experiments, we evaluate our models and the baselines under different partial observation functions. These observation functions are regulated by the random mask variables $m_{0:H}$, where $H$ represents the episode length. Here, $m_t^i = 1$ suggests that at time step $t$, the $i$-th state factor is not observed, otherwise, it is observed. In every test run, given the missing scenario and factor missing ratio $\eta$, we select this mask variable based on Algorithm 3. It's important to note that since the scenario is pre-set, there's no need to sample the variable $p$ to decide which scenario will be used.

**IQL baselines.** For the IQL_FA and IQL_FZ baselines, we employ the same hyper-parameters as those utilized in the training of the decision head (refer to Section A.2 for details). For the IQL_R baseline, we combine IQL [2] with a cutting-edge method for online RL in POMDPs [3] to construct this baseline. This strategy employs a unique recurrent neural network architecture to address partial observability and is compatible with any actor-critic algorithms. Specifically, we substitute the fully-connected networks in the IQL implementation with the recurrent architectures suggested in the method. We use the official implementation of the recurrent architecture for our experiments[1]. Following this, we employ a grid search strategy to finalize our choice of hyper-parameters, the details of which are reported in Table 5.

On the other hand, in our experiments, IQL_R underperforms in all scenarios when observation functions are diverse and uncertain. However, it's noteworthy that IQL_R fares well in settings where the observation functions are singular and stable. Specifically, we use two commonly adopted observation functions on locomotion tasks [3]. First, we mask the position information of robots (denoted as **Mask-P**), and second, we mask the velocity information of robots (denoted as **Mask-V**). We provide the average testing performance in Table 5. These results indicate that while IQL_R

---

[1]https://github.com/twni2016/pomdp-baselines

struggles to formulate effective policies under varied and dynamic partial observation settings, it can perform well when the observation function is singular and fixed in the offline setting. A possible explanation for this is that the diverse partial observation function leads to enhanced non-stationarity of dynamics, which makes direct policy training on these cases challenging and often results in highly unstable training.