# OpenReview forum: "Offline RL with Discrete Proxy Representations for Generalizability in POMDPs"
_NeurIPS.cc/2023/Conference — NeurIPS 2023 poster_

### Official Review · Reviewer_D69B · 2023-06-19

**Soundness:** 3 good
**Presentation:** 3 good
**Contribution:** 3 good
**Rating:** 6
**Confidence:** 4

**Summary:**

This paper proposes ORDER, an offline RL solution where the agent have access to states in the offline data, but can only witness observations where some groups of dimensions are masked when deployed due to occlusion or perturbation in real-life scenarios. To address this problem, a 3-step solution is proposed. First, a state encoder is trained to convert the states into discrete representations, which is essentially multiple VQ-VAEs grouped by observability. Then, the RL agent is trained with IQL on the offline dataset with state dimensions randomly blocked, using representations produced by the encoder as observations. Finally, a proxy state encoder is trained to estimate the current discrete representation based on the past trajectory and current state in deploy time, and its output is used for the trained RL agent. On several mujoco testbeds, ORDER is proved to work better than the baselines and gives representation with better alignment to ground truth.

**Strengths:**

1.	This paper is clearly written and easy to follow. Figure 1, 2 and 3 effectively summarizes the main idea of the algorithm, helping the reader to understand. The math symbols are organized and the motivation for each component such as the use of discrete representations are well-explained.

2.	The solution proposed by this paper is sound, novel and makes much sense, especially the discrete representation part. From my understanding, there could be two advantages of using discrete representation over continuous: 1) the training of VQ-VAE automatically encourages different values of states to be clustered into one representation, and thus becoming more robust to estimation by the proxy representation, and 2) the RL agent works better on the discrete state space.

3.	The experimental results is neat and clearly shows the superiority of ORDER to other baselines; for example, the visualization result in Fig. 6 effectively shows why discrete representation is used.


**Weaknesses:**

**1.	The related work section could be improved.**

A)	The reference list at a single point is too long (line 53, 55 and 78), while the occurrence of reference is too few in the section. The referenced related work is not (only) a proof that authors have read many papers, but (also) should summarize the prior works with a taxonomy that helps the reader to understand the main lines of work and the position of this work in the context. For example, in line 78, “both empirical and theoretical studies …” should be “both empirical […] and theoretical […] studies …”; in line 93-97, point i) and ii) should each have a reference list; in line 103-104, a list of example should be made about “specific partial observation” with reference list for each kind.

B)	There should be a brief discussion on sim2real in the robotics community, since the motivation of the paper is to train the agent in a controlled laboratory environment (quoted from line 34) and delopyed in real-world disaster scenarios (quoted from line 37).

**2.	The experiment section could be stronger.**

A)	More clarification could be made about the second point in the strength. Specifically, is the performance better because of better representations, or better because IQL works better in the discrete state space?

B)	More ablation could be added. For example, what if multiple dimensions are in one group of visibility (I assume the hyperparameters for VQ-VAE would change)? What is the effect of the dimension and the number of discrete codes of the codebook of VQ-VAE? What if the dataset is not medium, but rather a mixture of expert and random?

**3.	The computational resource consumption, negative societal impact and licenses are not specified.**

**4.	Other minor problems:**

A)	Line 285, bseline -> baseline;

B)	Line 60 in the appendix, GRU should be cited;

C)	I would suggest the authors to give clearer, more intuitive explanation at the beginning of the paper about why discrete rather than continuous representation is used. Currently, the reason for using discrete representation is only stated at the end of introduction, and is only empirical without intuition.


**Questions:**

I have one question below, apart from those in the weakness section:

From line 196, we know that one encoder only takes one state factor into account; meanwhile, from line 14-15 in the appendix, we know that each dimension is seen as a state factor. Does this mean that the embedding is generated assuming that (groups of) dimensions are independent to each other? If it is, then there is a problem: the agent perceives the same environment, which should not be independent on different sources (with different mask of visibility).

Below are my suggestions for the author. I will be happy to increase my score if the authors address those problems (see weakness sections for details on point 1-4):

1. Improve the related work section;

2. Present more ablation results;

3. Append the computational resource consumption, negative societal impact and licenses;

4. Fix the other minor problems;

5. I strongly advice the authors to open source upon the acceptance (or next submission) of the paper.


**Limitations:**

The paper does a good job in discussing the limitations, and I am convinced that despite of those limitations, the work is still valuable to the RL community; however, the paper does not mention any potential negative societal impact of the work. For example, while mujoco is still far from real-life applications, works on automated decision making from lab environment to real life is possible to substitute human work and cause job loss.

---

> ### Author Rebuttal · Authors · 2023-08-10
>
> Dear Reviewer:
>
> Thank you for your thoughtful questions and feedback on our paper. Below, we provide detailed responses to each of your questions:
>
> # Q1. The embedding is generated assuming that (groups of) dimensions are independent to each other?
>
> While our model considers individual state factors as described in line 196, it doesn't inherently imply independence across these dimensions. Here's why:
>
> - **Embedding Concatenation:** Referencing Equation 2 and Equation 4, the embedding concatenation mechanism amalgamates information across all dimensions. This ensures integrated representation and counters the notion of isolated dimensions.
>
> - **Trajectory Encoder Integration:** As highlighted by Equation 5 and elaborated in line 252, our trajectory encoder, denoted as \(\xi\), incorporates historical data from all dimensions when forming current proxy representations. This further solidifies the interdependence among dimensions.
>
> We hope this provides clarity regarding our model's treatment of state dimensions. The interconnectedness of dimensions, ensured by our design choices, aptly captures the essence of agents perceiving a unified environment, regardless of mask variations.
>
>
> # Improvements to the Related Work Section
>
> **A. Organization of References:**
>
> We value your constructive feedback on refining our Related Work section. To enhance the readability and utility of our related work section:
>
> 1. **Reference Consolidation:** We will reduce long lists of references at a single point, ensuring that they are spread more evenly and are contextually relevant.
> 2. **Enhanced Summarization:** We aim to provide a more structured summary of prior works, incorporating a taxonomy that facilitates a clearer understanding of the existing literature landscape.
>
> **B. Discussion on Sim2Real:**
>
> The term "sim2real" embodies the transition from models and algorithms optimized in a simulated environment to their application in real-world settings. This transition is particularly vital in robotics and automated systems where real-world conditions can be unpredictable and varied. In the context of our paper, while we predominantly focus on simulated environments, the end goal is to ensure that our findings are not just theoretically sound but also practically viable in real-world scenarios. Understanding and bridging the "sim2real" gap is a crucial step in this direction. We will incorporate a dedicated subsection that discusses prominent contributions from the robotics community on "sim2real". This will better position our work in the broader context and underline its relevance.
>
> # Improvements to the Experiment Section
>
> Thank you for highlighting the need for clarity on the strengths of our discrete representation approach.
>
> The improved performance is attributed to two key factors:
>    -  VQ-VAE's training inherently promotes clustering of diverse state values into a unified representation, enhancing the robustness of the proxy representation.
>    - A discrete state space, being more constrained than its continuous counterpart, benefits offline RL algorithms. Such a setup increases the likelihood of encountering similar states during training, thereby mitigating the out-of-distribution (o.o.d.) challenge.
>
> We'll ensure that the experiment section further elucidates these distinctions. Your feedback is pivotal in sharpening our paper's clarity, and we're grateful for it.
>
> # Additional Ablation Studies
>
> Due to time constraints and space limitations, we couldn't present new ablation results currently. However, we're committed to providing more comprehensive results in the revision:
>    1) Ablations for cases where multiple dimensions fall within a single group.
>    2) Ablation of the dimensionality of the discrete code (note that variations in the number of discrete codes have already been showcased in Figure 6 (d)).
>    3) Expanded results across different datasets, including expert, medium-replay, and medium-expert datasets.
>
> # Computational Resources and Timeframes
>
> All experiments were executed on Ubuntu 20.04, utilizing an Nvidia Tesla A6000 GPU. On average, training on a dataset required 2732.63 seconds, while inference for one episode (with a maximum of 1000 timesteps) took about 8.17 seconds. We will offer a detailed breakdown of these costs for each task in our revised submission.
>
> # Societal Impact Discussion
>
> While our research is grounded in the controlled Mujoco environment, the implications of transferring these findings to real-world settings merit consideration. Automated decision-making systems, if universally adopted, might reduce human roles in certain sectors, potentially leading to job displacements. It's crucial to strike a balance, ensuring technological advancements align with societal welfare, and to approach real-world deployments with an awareness of these broader implications.
>
> # Addressing Minor Concerns
>
> We'll correct the typo on Line 285 and add a GRU citation on Line 60 in the appendix.
>
>    - **Clarification on Discrete Representation**:  In the revised version, we'll provide a more intuitive explanation early in the introduction: Imagine the difference between sorting colors into broad categories versus matching exact shades. The former, like a discrete state space, makes generalization easier by grouping similar items together. This is beneficial for offline RL algorithms, as it heightens the chance of re-encountering similar states during training, effectively addressing the out-of-distribution (o.o.d.) challenge inherent in continuous spaces.
>
> # Commitment to Code Release
>
> We will release our implementation after acceptance.

---

> > ### Comment · Reviewer_D69B · 2023-08-11
> > **Response to the Rebuttal**
> >
> > Thanks for the detailed response; I appreciate the authors' effort to address my concerns, and I think concerns like intuitions of using discrete  representation are well-addressed. I would keep my score for now, however, as the ablations are not provided (especially multiple dimensions fall within a single group, which is also requested by other reviewers such as iLBR).

---

> > > ### Author Response · Authors · 2023-08-15
> > > **A Response for Additional Ablation Studies (1/3)**
> > >
> > > Dear Reviewer,
> > >
> > > Thank you for your constructive feedback.
> > >
> > > In response, we have conducted additional experiments and present new ablation results that further substantiate our findings. We have outlined these results below:
> > >
> > > # Ablation for Cases with Multiple Dimensions in a Single Group
> > > In this ablation study, we grouped multiple dimensions into a single entity, using a state factor encoder to process the information within this group. When the remaining number of dimensions is smaller than the specified group size, we treated these dimensions as a separate group, even if their count is less than the group size. Our findings reveal that our model maintains superior performance even when the group size exceeds one.
> > >
> > > __Table 1: Generalization performance under dynamic missing scenarios when multiple dimensions fall within a single group with different missing rations $\eta$ (%).__
> > >
> > > |Group size| Dataset | Method | $\eta=10\%$ | $\eta=30\%$ | $\eta=50\%$ | $\eta=70\%$ |$\eta=90\%$ |
> > > | :---: | :---: | :---: | :---: | :---: | :---: | :---: |:---: |
> > > 2 |halfcheetah-medium-v2| IQL_ORDER| __41.5__| __38.2__|__37.4__|__34.3__|__27.6__|
> > > 2 |halfcheetah-medium-v2| IQL_R| 7.5| 6.5|3.4|7.3|2.6|
> > > 2 |halfcheetah-medium-v2| IQL_FA|  31.5| 19.5|16.4|10.3|9.6|
> > > 2 |halfcheetah-medium-v2| IQL_FZ|  30.5| 16.5|16.4|8.3|6.4|
> > > 2 |hopper-medium-v2| IQL_ORDER| __72.5__| __70.1__|__62.2__|__57.4__|__50.6__|
> > > 2 |hopper-medium-v2| IQL_R| 20.5| 16.2|12.4|17.3|4.9|
> > > 2 |hopper-medium-v2| IQL_FA|  40.2| 36.2|22.4|19.3|7.4|
> > > 2 |hopper-medium-v2| IQL_FZ|  19.5| 16.3|12.4|11.3|6.7|
> > > 2 |walker2d-medium-v2| IQL_ORDER| __53.4__| __52.5__|__43.4__|__30.2__|__11.6__|
> > > 2 |walker2d-medium-v2| IQL_R| 5.5| 6.2|4.3|7.2|2.6|
> > > 2 |walker2d-medium-v2| IQL_FA|  50.2| 46.3|32.1|__30.7__|9.6|
> > > 2 |walker2d-medium-v2| IQL_FZ|  10.4| 6.5|6.4|7.3|4.2|
> > >
> > > |Group size| Dataset | Method | $\eta=10\%$ | $\eta=30\%$ | $\eta=50\%$ | $\eta=70\%$ |$\eta=90\%$ |
> > > | :---: | :---: | :---: | :---: | :---: | :---: | :---: |:---: |
> > > 3 |halfcheetah-medium-v2| IQL_ORDER| __40.2__| __36.2__|__32.2__|__24.0__|__21.3__|
> > > 3 |halfcheetah-medium-v2| IQL_R| 8.4| 4.2|3.7|2.4|2.0|
> > > 3 |halfcheetah-medium-v2| IQL_FA|  31.5| 17.5|13.4|12.2|8.7|
> > > 3 |halfcheetah-medium-v2| IQL_FZ|  21.3| 16.6|13.5|12.3|6.3|
> > > 3 |hopper-medium-v2| IQL_ORDER| __63.1__| __60.2__|__52.4__|__47.3__|__40.5__|
> > > 3 |hopper-medium-v2| IQL_R| 12.3| 11.3|5.3|7.0|4.7|
> > > 3 |hopper-medium-v2| IQL_FA|  37.3| 32.1|21.3|14.3|4.4|
> > > 3 |hopper-medium-v2| IQL_FZ|  19.3| 12.1|12.0|11.0|3.3|
> > > 3 |walker2d-medium-v2| IQL_ORDER| __52.3__| __42.1__|__33.2__|__27.2__|__10.2__|
> > > 3 |walker2d-medium-v2| IQL_R| 4.3| 2.1|4.2|4.0|2.1|
> > > 3 |walker2d-medium-v2| IQL_FA|  34.2| 32.1|23.4|21.7|9.4|
> > > 3 |walker2d-medium-v2| IQL_FZ|  10.9| 4.1|5.1|7.1|4.9|
> > >
> > > __Table 2: Generalization performance under factor reduction scenarios when multiple dimensions fall within a single group with different missing rations $\eta$ (%).__
> > >
> > > |Group size| Dataset | Method | $\eta=10\%$ | $\eta=30\%$ | $\eta=50\%$ | $\eta=70\%$ |$\eta=90\%$ |
> > > | :---: | :---: | :---: | :---: | :---: | :---: | :---: |:---: |
> > > 2 |halfcheetah-medium-v2| IQL_ORDER| __41.3__| __37.5__|__31.4__|__21.4__|__11.6__|
> > > 2 |halfcheetah-medium-v2| IQL_R| 10.3| 6.2|4.7|7.8|6.5|
> > > 2 |halfcheetah-medium-v2| IQL_FZ|  10.9| 6.7|2.1|2.3|2.8|
> > > 2 |hopper-medium-v2| IQL_ORDER| __65.5__| __66.1__|__41.3__|__21.2__|__13.2__|
> > > 2 |hopper-medium-v2| IQL_R| 7.5| 6.3|4.9|7.0|2.1|
> > > 2 |hopper-medium-v2| IQL_FZ|  12.5| 6.2|5.4|7.1|6.9|
> > > 2 |walker2d-medium-v2| IQL_ORDER| __53.6__| __26.3__|__22.1__|__14.2__|__6.3__|
> > > 2 |walker2d-medium-v2| IQL_R| 3.5| 7.4|2.3|2.1|6.7|
> > > 2 |walker2d-medium-v2| IQL_FZ|  43.5| 21.3|12.0|7.3|8.7|
> > >
> > > |Group size| Dataset | Method | $\eta=10\%$ | $\eta=30\%$ | $\eta=50\%$ | $\eta=70\%$ |$\eta=90\%$ |
> > > | :---: | :---: | :---: | :---: | :---: | :---: | :---: |:---: |
> > > 3 |halfcheetah-medium-v2| IQL_ORDER| __41.7__| __32.1__|__29.1__|__21.7__|__10.7__|
> > > 3 |halfcheetah-medium-v2| IQL_R| 10.9| 8.1|7.9|7.1|5.4|
> > > 3 |halfcheetah-medium-v2| IQL_FZ|  10.1| 6.9|3.2|3.4|1.9|
> > > 3 |hopper-medium-v2| IQL_ORDER| __63.1__| __52.2__|__42.1__|__32.1__|__13.1__|
> > > 3 |hopper-medium-v2| IQL_R| 7.4| 5.2|4.0|9.5|3.2|
> > > 3 |hopper-medium-v2| IQL_FZ|  15.1| 6.1|3.2|8.2|7.3|
> > > 3 |walker2d-medium-v2| IQL_ORDER| __54.2__| __27.1__|__23.2__|__12.1__|__7.4__|
> > > 3 |walker2d-medium-v2| IQL_R| 3.8| 7.9|2.9|2.2|2.0|
> > > 3 |walker2d-medium-v2| IQL_FZ|  41.0| 22.4|22.1|21.4|23.7|

---

> > > ### Author Response · Authors · 2023-08-15
> > > **A Response for Additional Ablation Studies (2/3)**
> > >
> > > # Ablation of the Discrete Code Dimensionality
> > > We have also explored how changes to the dimension or number of discrete codes impact our model's performance. We present the averaged scores of our model in the 'hopper-medium-v2' dataset as evidence. Our results show that increasing the number of discrete codes generally leads to improved model performance, likely due to an enhanced expressive capacity. However, beyond 40 codes, the gains in performance become negligible. This suggests an optimal trade-off between the number of discrete codes and model performance. Additionally, our experiments show that a code dimension of 2 yields excellent performance, but further increasing this parameter does not yield significant improvement.
> > >
> > > __Table 3: Ablation results. Averaged score of different number of discrete codes in the hopper-medium-v2 dataset.__
> > >
> > > |# discrete codes| Averaged score|
> > > | :---: | :---: |
> > > |10|40.2|
> > > |20|56.3|
> > > |30|49.5|
> > > |40|60.2|
> > > |50|62.3|
> > > |60|62.4|
> > >
> > > __Table 4: Ablation results. Averaged score of different codebook dimension in the hopper-medium-v2 dataset.__
> > >
> > > |# codebook dimension| Averaged score|
> > > | :---: | :---: |
> > > |1|55.3|
> > > |2|60.2|
> > > |4|60.7|
> > > |8|59.2|
> > > |16|56.9|
> > > |32|58.2|
> > >
> > > # Expanded Results Across Different Datasets
> > > To further validate our model’s generalizability, we have extended our experiments to include additional datasets—namely 'expert', 'medium-replay', and 'medium-expert'. Our extended results confirm that, in most cases, our model outperforms other baseline approaches across these varied datasets.
> > >
> > >
> > > __Table 5: Generalization performance of different methods under dynamic missing scenarios  with different missing rations $\eta$ (%).__
> > >
> > > | Dataset | Method | $\eta=10\%$ | $\eta=30\%$ | $\eta=50\%$ | $\eta=70\%$ |$\eta=90\%$ |
> > > | :------------------------: | :----------: | :---: | :---: | :---: | :---: |:---: |
> > > |halfcheetah-medium-expert-v2| IQL_ORDER| 80.7| __76.3__|__70.4__|__67.3__|__60.6__|
> > > |halfcheetah-medium-expert-v2| IQL_R| 22.4| 20.3|17.4|15.6|13.2|
> > > |halfcheetah-medium-expert-v2| IQL_FA| __80.9__| 63.3|53.5|30.2|20.4|
> > > |halfcheetah-medium-expert-v2| IQL_FZ| 77.3| 52.1|32.2|21.2|18.4|
> > > |halfcheetah-medium-replay-v2| IQL_ORDER| __40.2__| __38.1__|__33.2__|__25.6__|__20.2__|
> > > |halfcheetah-medium-replay-v2| IQL_R| 10.2| 7.1|5.2|6.1|9.0|
> > > |halfcheetah-medium-replay-v2| IQL_FA| 37.2| 33.2|26.2|18.4|10.5|
> > > |halfcheetah-medium-replay-v2| IQL_FZ| 38.6| 23.5|15.2|12.4|8.3|
> > > |halfcheetah-expert-v2| IQL_ORDER| __90.8__| __83.5__|__80.2__|__67.3__|__52.1__|
> > > |halfcheetah-expert-v2| IQL_R| 26.4| 22.5|17.2|19.6|15.7|
> > > |halfcheetah-expert-v2| IQL_FA| 88.3|80.4|75.1| 65.0|40.9|
> > > |halfcheetah-expert-v2| IQL_FZ| 80.3|78.2|71.2| 60.0|42.1|
> > >
> > > | Dataset | Method | $\eta=10\%$ | $\eta=30\%$ | $\eta=50\%$ | $\eta=70\%$ |$\eta=90\%$ |
> > > | :------------------------: | :----------: | :---: | :---: | :---: | :---: |:---: |
> > > |hopper-medium-expert-v2| IQL_ORDER| __87.7__| __77.2__|__70.0__|__65.2__|__61.0__|
> > > |hopper-medium-expert-v2| IQL_R| 12.4| 20.5|10.4|9.7|12.3|
> > > |hopper-medium-expert-v2| IQL_FA| 86.7| 63.6|57.3|35.3|23.8|
> > > |hopper-medium-expert-v2| IQL_FZ| 85.3| 55.3|36.4|26.2|22.6|
> > > |hopper-medium-replay-v2| IQL_ORDER| __90.2__| __88.3__|__83.6__|__65.6__|__50.2__|
> > > |hopper-medium-replay-v2| IQL_R| 17.4| 13.5|13.4|10.0|9.5|
> > > |hopper-medium-replay-v2| IQL_FA| 88.8| 83.6|77.5|55.3|23.8|
> > > |hopper-medium-replay-v2| IQL_FZ| 86.8| 73.6|67.4|35.3|18.2|
> > > |hopper-expert-v2| IQL_ORDER| __91.8__| __84.3__|__75.2__|__66.2__|__55.7__|
> > > |hopper-expert-v2| IQL_R| 15.4| 12.5|14.7|8.3|8.5|
> > > |hopper-expert-v2| IQL_FA| 89.9| 80.6|72.0|51.2|34.2|
> > > |hopper-expert-v2| IQL_FZ| 88.3| 78.4|64.2|41.3|20.6|
> > >
> > > | Dataset | Method | $\eta=10\%$ | $\eta=30\%$ | $\eta=50\%$ | $\eta=70\%$ |$\eta=90\%$ |
> > > | :------------------------: | :----------: | :---: | :---: | :---: | :---: |:---: |
> > > |walker2d-medium-expert-v2| IQL_ORDER| __100.2__| __96.5__|__82.4__|__77.5__|__66.6__|
> > > |walker2d-medium-expert-v2| IQL_R| 31.5| 24.3|14.4|25.4|23.1|
> > > |walker2d-medium-expert-v2| IQL_FA| 95.9| 83.3|63.5|50.6|40.4|
> > > |walker2d-medium-expert-v2| IQL_FZ| 97.3| 62.1|52.4|46.2|28.3|
> > > |walker2d-medium-replay-v2| IQL_ORDER| 66.1| __58.2__|__53.2__|__45.6__|__30.1__|
> > > |walker2d-medium-replay-v2| IQL_R| 8.2| 9.1|5.6|7.3|6.0|
> > > |walker2d-medium-replay-v2| IQL_FA| __67.2__| 43.1|36.3|28.4|10.0|
> > > |walker2d-medium-replay-v2| IQL_FZ| 66.6| 33.0|25.2|22.7|12.5|
> > > |walker2d-expert-v2| IQL_ORDER| __106.2__| __93.3__|__78.2__|__73.2__|__60.4__|
> > > |walker2d-expert-v2| IQL_R| 46.4| 42.3|37.1|29.5|25.7|
> > > |walker2d-expert-v2| IQL_FA| 98.3|90.4|55.1| 35.2|30.5|
> > > |walker2d-expert-v2| IQL_FZ| 100.3|78.4|37.2| 33.0|22.1|

---

> > > ### Author Response · Authors · 2023-08-15
> > > **A Response for Additional Ablation Studies (3/3)**
> > >
> > > __Table 6: Generalization performance of different methods under factor reduction scenarios  with different missing rations $\eta$ (%).__
> > >
> > > | Dataset | Method | $\eta=10\%$ | $\eta=30\%$ | $\eta=50\%$ | $\eta=70\%$ |$\eta=90\%$ |
> > > | :------------------------: | :----------: | :---: | :---: | :---: | :---: |:---: |
> > > |halfcheetah-medium-expert-v2| IQL_ORDER| __82.1__| __77.3__|__55.4__|__27.3__|__20.5__|
> > > |halfcheetah-medium-expert-v2| IQL_R| 20.0| 18.3|17.2|10.6|13.9|
> > > |halfcheetah-medium-expert-v2| IQL_FZ| 75.5| 30.4|22.2|20.2|14.3|
> > > |halfcheetah-medium-replay-v2| IQL_ORDER| __39.4__| __35.1__|__21.5__|__15.3__|__10.2__|
> > > |halfcheetah-medium-replay-v2| IQL_R| 10.0| 11.3|6.3|9.1|10.2|
> > > |halfcheetah-medium-replay-v2| IQL_FZ| 33.6| 23.3|15.4|10.4|11.3|
> > > |halfcheetah-expert-v2| IQL_ORDER| __89.5__| __83.3__|__67.2__|__47.3__|__32.0__|
> > > |halfcheetah-expert-v2| IQL_R| 19.4| 12.4|10.3|10.3|13.5|
> > > |halfcheetah-expert-v2| IQL_FZ| 80.5|48.1|31.3| 20.0|22.3|
> > >
> > > | Dataset | Method | $\eta=10\%$ | $\eta=30\%$ | $\eta=50\%$ | $\eta=70\%$ |$\eta=90\%$ |
> > > | :------------------------: | :----------: | :---: | :---: | :---: | :---: |:---: |
> > > |hopper-medium-expert-v2| IQL_ORDER| __86.4__| __73.1__|__50.4__|__35.1__|11.0|
> > > |hopper-medium-expert-v2| IQL_R| 12.4| 20.5|10.4|9.7|9.3|
> > > |hopper-medium-expert-v2| IQL_FZ| 85.3| 25.3|16.3|16.2|__12.5__|
> > > |hopper-medium-replay-v2| IQL_ORDER| __91.7__| __78.2__|__53.2__|__35.3__|__20.1__|
> > > |hopper-medium-replay-v2| IQL_R| 19.4| 14.5|12.3|12.0|10.3|
> > > |hopper-medium-replay-v2| IQL_FZ| 89.8| 33.6|17.3|15.2|13.5|
> > > |hopper-expert-v2| IQL_ORDER| __90.3__| __74.2__|__55.1__|__26.1__|__15.5__|
> > > |hopper-expert-v2| IQL_R| 12.4| 14.5|18.2|8.3|11.3|
> > > |hopper-expert-v2| IQL_FZ| 84.3| 22.4|14.1|11.3|10.9|
> > >
> > > | Dataset | Method | $\eta=10\%$ | $\eta=30\%$ | $\eta=50\%$ | $\eta=70\%$ |$\eta=90\%$ |
> > > | :------------------------: | :----------: | :---: | :---: | :---: | :---: |:---: |
> > > |walker2d-medium-expert-v2| IQL_ORDER| __100.5__| __76.5__|__42.4__|__27.3__|__6.6__|
> > > |walker2d-medium-expert-v2| IQL_R| 11.5| 4.3|4.5|5.9|3.1|
> > > |walker2d-medium-expert-v2| IQL_FZ| 97.3| 62.1|52.4|46.2|28.3|
> > > |walker2d-medium-replay-v2| IQL_ORDER| __63.1__| __50.2__|__33.1__|__25.9__|__10.1__|
> > > |walker2d-medium-replay-v2| IQL_R| 8.1| 8.5|10.6|4.2|8.0|
> > > |walker2d-medium-replay-v2| IQL_FZ| 62.6| 23.1|15.3|12.2|8.5|
> > > |walker2d-expert-v2| IQL_ORDER| __103.2__| __73.3__|__48.1__|__23.2__|5.4|
> > > |walker2d-expert-v2| IQL_R| 16.4| 14.4|17.3|9.2|6.7|
> > > |walker2d-expert-v2| IQL_FZ| 81.2|58.4|27.1| 13.0|__12.2__|
> > >
> > >
> > >
> > > We hope these additional experiments and clarifications address your concerns and demonstrate the robustness and efficacy of our approach.
> > >
> > > Thank you once again for your valuable insights.
> > >
> > > Sincerely,
> > > Authors

---

> > > > ### Comment · Reviewer_D69B · 2023-08-15
> > > > **Response**
> > > >
> > > > Thanks for your detailed experiment results. I think my concerns are addressed and thus I will raise my score from 5 to 6.

---

> > > > > ### Author Response · Authors · 2023-08-18
> > > > >
> > > > > Thank you for rasing the score. We appreciate your valuable feedback on improving our work. If you have any other questions, please post them and we are happy to continue our communication.

---

### Official Review · Reviewer_AAho · 2023-06-29

**Soundness:** 2 fair
**Presentation:** 2 fair
**Contribution:** 1 poor
**Rating:** 4
**Confidence:** 4

**Summary:**

The work looks to tackle a subset of the partially observable offline reinforcement learning problem setting, where a dataset of offline experience (of full states and masked states) is given during training, and an agent is tested on masked state features during test time. The authors propose the ORDER training framework, where during training time, an agent first learns discrete state representations using the full states, then uses an RNN to learn (in a supervised manner from the discrete representations from the full states) the discrete state representations over the masked states.

With this framework in place, the authors show that this approach does well in a masked version of the D4RL offline reinforcement learning benchmark, which they create themselves. They show this form of discretization outperforms baselines algorithm performs better on this benchmark as compared to other baselines. They also show an improvement as a higher percentage of the states are masked out.

**Strengths:**

Within this specific problem setting, the authors do a good job of showing the benefits of their framework and of discretization. It seems that with this problem set up, discretization and supervised learning over masked states helps quite a bit with performance, and is also quite robust to the percentage of masking based on the plots shown in section 5.

**Weaknesses:**

There are a few important issues that this paper seems to have, which I’ll describe in broad strokes in this section. For more specific details on these issues, please see the section-by-section review.

The first big issue is that it seems that the work has mixed their problem setting with their solution method. The two seem innately tied - even the structure of the work mixes the two together. While the authors start the work with describing the general problem setting of partial observability in offline reinforcement learning, the work continues on to describe a problem setting that is quite far off that mark - where the partial observability only stems from simple bernoulli masks, and full state information is given during the training phase. Besides this, additional assumptions are also given to the agent, such as the agent actually knowing *****which***** state features are masked when (which means that **************************************************************************the agent has to know what’s wrong with the state features at every step)**************************************************************************. All of these assumptions on the problem setting are necessary for ORDER to work. While I’m not saying there’s anything fundamentally wrong with the problem setting they describe, **the issue arises with the fact that the proposed solution method, based on the evidence presented in this work, seems to only work in this extremely specific problem setting that they’ve introduced.**

The language used and structure throughout the work are also quite misleading, especially in the introduction and abstract where it seems the work has overclaimed in the introduction and underdelivered from section 3 onwards. It feels like before reading section 3, you describe your method as general with the ability to tackle POMDPs in an offline setting. But after reading your preliminary section, you seem to reduce the scope suddenly (masked observation functions, different observation functions, access to states during training). I would recommend adding specificity to the problem setting that you describe throughout your work.

Besides this, I find it somewhat suspicious that IQL_R doesn’t seem to work in any of the environments/datasets, and shows the same flat line throughout all environments/datasets. This calls into question implementation or how hyperparameter tuning was done, which was lacking in the appendix.

Lastly, I’m also surprised you didn’t reference hindsight state information, seeing as their problem setup is similar to yours. It seems that this setup is more similar to what the authors are trying to tackle.

**Questions:**

Here is a central question I would like answered with regards to the work: From Appendix B, “Following this, we employ a grid search strategy to finalize our choice of hyper-parameters, the details of which are reported in Table 5.” What was this grid search strategy? How did you arrive at these hyperparameters?

Besides this question, here is a section-by-section review/questions with regard to the work:

****1.****

“While POMDP methods excel in online RL, they rely on continuous environment interactions for policy adaptation, making them unsuitable for offline RL” - This statement confused me a bit. Do you mean that you need on-policy samples for POMDP methods?

“we aim to tackle a more general POMDP problem” - This is false. Your problem set up is not more general, seeing as you have access to states in training. The most general class of POMDP problem settings are settings where you don’t have access to state at all. This setting encompasses “POMDP families” by simply expanding the state space instead of saying that there are multiple observation functions.

This formulation and claim that the authors are making about tackling a “more general POMDP problem” is also **untrue** in your problem setting. There aren’t different observation functions between train and test time. You simply use a stochastic observation function.

****3.****

“In particular, during the training period, the policy has access to the underlying MDP” - I’m assuming you mean *******samples which include state information*******? If you have the underlying MDP then why not do online RL? Please be more precise.

****4.****

“By converting unseen partial observations into discrete forms akin to those present in the training data”

**Is all these machine learning machinery necessary to discretize states?** We’ve had tile coding and different, very effective, state discretization schemes for a while now. Why do we need a VQ-VAE?

“And we introduce partial observability by randomly setting a portion of mask variables m^i_t to be 0, simulating real-world partially observable scenarios.” - this entire problem set up is a big red flag. **The problem setup is not a realistic representation of partial observability ******at all********. From reading this paper, it sounds like you’ve essentially worked backwards, where you’ve defined a constrained solution method first, then defined your problem setting based on this solution method. It seems like your solution method essentially hinges on the fact that your partial observations are of the form of *************independently************* masking your state feature vectors, which is almost never the case in the real world.

Do I understand this correctly that ************************************************************************************************************************the agent is given the masking vector M?********************************************************************************* If this is the case, then this is another huge assumption. The LSTM here is essentially just predicting missing state features, knowing which are missing.

**4.2.2**

What is g here?

****5.****

“Our results clearly indicate that ORDER substantially enhances the generalization performance of policies trained on offline datasets in diverse partially observable conditions.” This is claim is tenuous at best. How is this one partial observability setting considered “diverse observable conditions?”

Looking at your appendix, I’m not sure how you’ve swept hyperparameters and arrived at the hyperparameters given for the baselines. Most importantly, the IQL_R baseline. I’m also unsure as to what IQL_R is exactly. Is it simply IQL with an RNN trained over the masked D4RL datasets?

******5.1******

“An intrinsic characteristic of a policy with strong generalization ability is its ability to maintain performance as the missing ratio increases.” - what does strong generalization performance mean? It seems you mean “generalization in terms of number of masked features”, but more specificity in this language is needed, seeing as “generalization” means a whole deluge of different things in the reinforcement learning context. **This seems to also be an issue throughout this work, seeing the context in which generalization is used.**

“This suggests that it struggles to develop effective policies under diverse and dynamic partial observation settings.” - This is markedly untrue in general - it might be the case where **************************************specifically in your problem setting that you invented************************************** this might be true, but I’m also unconvinced this is the case currently as I’m unsure of how you’ve swept and decided on hyperparameters.

******5.2******

Is using the overlap in t-SNE the best idea for seeing overlap in representations? It’s a dimensionality reduction technique, so the overlap doesn’t necessarily mean much. Why not just use a cosine similarity metric instead? It would be better to have numbers to show the “similarities” in the representations.

****6.****

“that addresses the challenges of partial observability in real-world scenarios.” - **this is just flat-out untrue. Mujoco with random Bernoulli masks is far from real-world scenarios.**

**Limitations:**

Please see weaknesses and questions.

---

> ### Author Rebuttal · Authors · 2023-08-10
>
> Dear Reviewer:
>
> Thank you for your thoughtful questions and feedback on our paper. Below, we provide detailed responses to each of your questions:
>
> # Clarifying Our Problem Setting and Its Real-World Relevance
>
> **Real-world Motivations:** Our specific assumptions, though tailored, are rooted in real scenarios where observation functions are diverse and often unpredictable. Our goal is to bridge this gap with robust solutions.
>
> - **Search and Rescue:** A robot, trained on complete offline data in controlled conditions, may face varied challenges when deployed in different disaster zones. Factors like smoke, murky waters, or sensor noise introduce unpredictable observation functions, emphasizing the need for our approach.
>
> - **Financial Markets:** Algorithms trained on comprehensive historical data can encounter diverse market conditions where only specific subsets of indicators are reliable. The ever-changing nature of markets results in unpredictable observation functions, further underscoring our approach's relevance.
>
> **Assumption Specificity:** We recognize that our assumptions represent distinct scenarios but stand firm in their practical significance:
>
> - **Full State Information during Training:** Though this assumption is seemingly idealistic, asymmetric RL for POMDPs [1-2] and RL for hindsight observable MDPs [3] have effectively leveraged this for addressing challenges, e.g., self-driving vehicles [1].
>
> - **Employing Masks for Simulating Diverse Observation Functions:**
>   1. Masks effectively capture some real-world partial observable scenarios, such as robotics where sensor occlusions can be modeled as feature masks.
>   2. Traditional POMDPs, as described in [3-6], utilize a consistent observation function that maps the full state to a partial one, often masking a predetermined set of factors. In contrast, our approach introduces unpredictability by avoiding assumptions about which factors are masked during testing. This results in diverse and unpredictable mapping functions during evaluation.
>   3. To represent this diversity, we employ two specific scenarios (refer to lines 235-241). It's essential to note that **our generated observation functions are not equivalent to a straightforward  stochastic Bernoulli function.** Our practical method of sampling multiple mask observation functions is explained in lines 74-80 of the Appendix. The complex interplay of differing missing ratios, distinct scenarios, and variable masked sections surpasses what can be captured by a singular Bernoulli distribution, highlighting the need for robust generalization.
>
> - **Awareness of Mask Vectors:** While we recognize that our assumption may not hold universally, it's pragmatically valid in many contexts and aligns with existing research conventions.
>   1. Practical domains like robotics and network systems frequently allow the detection of 'missing' or anomalous data, evidenced by discernible sensor disturbances or notable anomaly indicators.
>   2. Our approach resonates with numerous established works in the POMDP realm [4] and studies addressing data missingness [7-8], which operate under similar assumptions regarding the awareness of mask vectors.
>
>
>
> # Addressing Concerns in Method Description: Proposed Revisions
>
> Thank you for feedback. We are committed to refining our paper:
> 1. We acknowledge discrepancies between our introduction, abstract, and Section 3. We will harmonize these parts to clearly depict our method's capabilities and constraints, ensuring the work's scope is evident from the start.
> 2.  We'll clarify the rationale behind our chosen test settings, including our decision to use masked observation functions, the diversity of observation functions, and the accessibility to states during training (as elaborated in the above response).
> 3.  For overarching clarity, we will comb through the manuscript ensuring the problem, method, and results' scope are consistently and lucidly presented.
>
>
>
>
> # Addressing IQL_R Performance and Hyperparameter Tuning Concerns
> We have provided a more detailed explanation, along with supplementary results and hyper-parameter selection tables, in the accompanying pdf file in the global response panel.
>
> # Distinguishing Our Method from Hindsight State Information [3]
> While there are similarities, key distinctions also exist. Our method investigates uncertain and diverse mask observation functions in an offline mode, in contrast to the online, singular observation function in hindsight state information. For clarity, we've summarized these differences in the Table 3 in the attached pdf:
>
> # Other Queries:
> - **Online POMDP in Offline RL:** Online RL is optimized for live interactions. Without adjustments, their direct application to offline RL can lead to poor performance [9].
> - **Access to MDP:** None. This will be clarified in the revision.
> - **Use of VQ-VAE:** VQ-VAE stands as SoTA in discrete representation learning, justifying its incorporation.
> - **'g' Defined:** Refers to the nearest lookup, as noted on line 198.
> - **Choosing t-SNE:**  According to [10], t-SNE ensures that similar objects align closely, while dissimilar ones are positioned farther apart, making it suitable for visual similarity comparisons.
>
>
>
> 1.  Robust asymmetric learning in pomdps. ICML 2021.
> 2.  Leveraging fully observable policies for learning under partial observability. CoRL 2022.
> 3.  Learning in POMDPs is Sample-Efficient with Hindsight Observability. ICML 2023.
> 4.  Deep recurrent belief propagation network for POMDPs. AAAI 2021.
> 5.  Deep variational reinforcement learning for pomdps. ICML 2018.
> 6.  Recurrent Model-Free RL Can Be a Strong Baseline for Many POMDPs. ICML 2022.
> 7.  GAIN: Missing Data Imputation using Generative Adversarial Nets. ICML 2018.
> 8.  Variational Selective Autoencoder: Learning from Partially-Observed Heterogeneous Data. AISTATS 2021.
> 9.  Off-Policy Deep Reinforcement Learning without Exploration. ICML 2019.
> 10.  Visualizing Data Using t-SNE. 2008

---

> > ### Comment · Reviewer_AAho · 2023-08-13
> > **Response**
> >
> > Thanks for the response. Although a few of my concerns have been addressed, my main concernshave still not been fully alleviated.
> >
> > **"Traditional POMDPs, as described in [3-6], utilize a consistent observation function that maps the full state to a partial one, often masking a predetermined set of factor"**
> >
> > While there is one work listed that does use random masking (4 in your given list), the other works introduce POMDPs that don't use masking as you've described it (or even random masking!).
> > Looking at classic work in the POMDP literature [1, 2], the partial observability present is much more complex in its nature. My main concern with this problem setting is the extensibility of this proposed solution method, especially since it relies so heavily on the masking assumptions made by the problem setting.
> >
> > **"We will harmonize these parts to clearly depict our method's capabilities and constraints, ensuring the work's scope is evident from the start."**
> >
> > Could you be more specific with what you'll be changing? My biggest issue with this point was the overclaiming in the first few sections of the work. What are you going to do to address these concerns of overclaiming?
> >
> > To conclude, while I do appreciate the effort in "ensuring the work's scope is evident from the start", my concerns with the problem setting are still largely unaddressed. I do not believe the problem setting is well motivated enough in its current form.
> >
> > [1] Leslie Pack Kaelbling, Michael L. Littman, Anthony R. Cassandra. Planning and acting in partially observable stochastic domains. Artificial Intelligence, Volume 101, Issues 1–2, 1998. Pages 99-134.
> >
> > [2] Michael L. Littman, Anthony R. Cassandra, Leslie Pack Kaelbling. Learning policies for partially observable environments: Scaling up.  Armand Prieditis, Stuart Russell. Machine Learning Proceedings 1995. Pages 362-370.

---

> > > ### Author Response · Authors · 2023-08-17
> > > **Responses for addressing your concerns ( Extensibility of Problem Setting )**
> > >
> > > Dear reviewer,
> > > Thank you for your timely and valuable feedback. I appreciate the opportunity to clarify the key aspects of our work.
> > >
> > > **Understanding of Problem Setting:** Firstly, I would like to clarify that our setting is designed to be versatile, not restricted to __explicit__ mask observation functions. Any partial observation scenario where certain state information is unobserved can be interpreted within our mask observation framework. We'll provide clarifying examples subsequently.
> > >
> > > **Relation to Classic POMDP Literature:** We recognize that our mask observation functions may not encompass all forms of partial observability, such as state perturbations or noises cited in classic POMDP works. However, we assert that our approach effectively models scenarios where certain state information is observable while others are not, which is a widespread and realistic form of partial observability.
> > >
> > > **Extensibility of Our Setting:** Our approach is crafted for broad applicability. It accommodates a wide array of mask observation functions, as identified in existing literature. To further substantiate the flexibility and relevance of our method, we will provide examples demonstrating its alignment with various established works.
> > >
> > >
> > >
> > >
> > > - **Example-1: Revisiting Frozen Lake** Referring to Section 6 of [1], the agent in the "Revisiting Frozen Lake" environment cannot observe the hazard's position. In our approach, the mask vector for the hazard is consistently set to 1, while other positions are set to 0. This implies the agent observes everything except the hidden hazard.
> > >
> > > - **Example-2: Safe Autonomous Vehicle Learning** Section 6 of [1] mentions that the "Safe Autonomous Vehicle Learning" environment's partial observability arises when the field of view is obstructed. In our model, obstructions are represented by setting the respective mask vector to 1.
> > >
> > > - **Example-3: LunarLander-P, LunarLander-V** In Section 5.1 of [2], the authors directly state:
> > >    >"Masking parts of the state to turn MDPs into POMDPs is common in previous work [34, 3, 35, 5], the agent only observes subsets of the full state."
> > >
> > >    This statement is in alignment with our approach, where unobserved state information is represented by setting corresponding mask vectors to 1.
> > >
> > > - **Example-4: Car-Flag** In Section 5.1 of [2], agents usually observe the car's position and velocity. However, proximity to the blue flag lets them observe the green flag's side (left/right). Our framework defaults the flag's side mask value to 1, changing it to 0 when observable.
> > >
> > > - **Example-5: 8 OpenAI Gym Locomotion Tasks** The "Experiments" section of [4] introduces partial observability to eight tasks by directly masking specific state information. These tasks include _HalfCheetah_, _Hopper_, _Ant_, _InvertedDoublePendulum_, _InvertedPendulum_, _Swimmer, Reacher, and Walker2d.
> > >
> > > - **Example-6: Flickering Atari** Section 5.2 of [5] employs the "Flickering Atari" as a POMDP benchmark, presenting a blank screen for half of the observations. Our methodology aligns perfectly with this: at each time step, our mask vector has a 50% chance of being set to 1, symbolizing a fully unobserved state.
> > >
> > >
> > > - **Example-7: Occlusion Benchmark in 8 'Standard POMDP' Environments** In subsection "Standard POMDP" of Section 5.1 of [6], VRM proposes the Occlusion Benchmark, comprising eight environments: _Hopper-P_, _Ant-P_, _Walker-P_, _Cheetah-P_, _Hopper-V_, _Ant-V_, _Walker-V_, _Cheetah-V_. Here, “-P” denotes observations of positions and angles only, and “-V” denotes observations of velocities only. In line with our approach, the mask vector for observable state information is set to 0, and for unobservable information, it is set to 1.
> > >
> > > - **Example-8: 4 'Meta RL' Environments** In subsection "Meta RL" of Section 5.1 of [6], environments _Semi-Circle_, _Wind_, _Cheetah-Dir_, and _Ant-Dir_ feature POMDPs where certain parameters in rewards or dynamics vary between episodes but remain constant within a single episode. In our framework, the mask vectors for these varying parameters are consistently set to 1, indicating that these parameters are not observable to the agent.
> > >
> > > - **Example-9:3 'Robust RL' Environments** In subsection "Robust RL" of Section 5.1 of [6], the environments _Cheetah-Robust_, _Hopper-Robust_, and _Walker-Robust_ are described. In these scenarios, certain hidden states, such as the density and friction coefficients of simulated robots, remain fixed throughout an episode. To simulate this partial observability, our approach involves masking these coefficients by setting their corresponding mask vectors to 1.
> > >
> > > ---

---

> > > ### Author Response · Authors · 2023-08-17
> > > **Responses for addressing your concerns ( Revisions for Ensuring the Work's Scope is Evident from the Start )**
> > >
> > > We understand the importance of precisely defining the scope and contributions of our work, and we are committed to addressing this in the revised manuscript. Here is a detailed breakdown of the specific changes we plan to make:
> > >
> > >
> > > 1. **Abstract Revision**:
> > >   We will revise the abstract to highlight the two key factors of our method:
> > >    - i) learning from __full observations__ during offline training, and
> > >    - ii) its designed application towards __masked partial observabilities__.
> > >    **Revised Abstract:**
> > >    > ...which brings crucial challenges of the deployment of offline RL methods: i) the policy trained on data with __full observability__ is not robust against the __masked partial observability__ during execution, and ii) the modality of the __masked partial observability__ is usually unknown during training. In order to address these challenges, we present Offline RL with Discrete pRoxy rEpresentations (ORDER), a probabilistic framework which leverages novel state representations to improve the robustness against diverse __masked partial observabilities__. Specifically, we propose a discrete representation of the states and use a proxy representation to recover the states from __masked partial observable trajectories__. The training of ORDER can be compactly described as the following three steps. i) Learning the discrete state representations on data with __full observations__, ii) Training the decision module based on the discrete representations, and iii) Training the proxy discrete representations on the data with various __masked partial observations__, aligning with the discrete representations. We conduct extensive experiments to evaluate ORDER, showcasing its effectiveness in offline RL for diverse __masked partially observable scenarios__ and highlighting the significance of discrete proxy representations in improved generalization performance.
> > > ORDER is a flexible framework to employ any state-of-the-art offline RL algorithms and we hope that ORDER can pave the way for the deployment of RL policy against various __masked partial observabilities__ in the real world.
> > >
> > >
> > > **2. Introduction Revision:**
> > >    - **Adding a Statement for Defining the Scope of Our Specific Problem Setting**
> > >    Revised statement to be inserted at the start of line 43 of the introduction section:
> > >    > Guided by this motivation, our work targets the specific challenge where, although complete state information is available from an offline dataset, the deployment stage might face various masked partial observations in which certain state information remains hidden, while other parts are visible
> > >
> > >    - **Highlighting the Limitations**
> > >    Revised limitation statement to be added in line 72:
> > >    > Nevertheless, it is important to acknowledge existing limitations. ORDER is presently tailored to address masked partial observation functions and does not extend to other forms, such as perturbations and noises. Additionally, the framework’s efficacy in complex real-world applications, like autonomous driving, warrants further investigation. Regardless, ORDER is a flexible framework...
> > >
> > >
> > > **3. Related Works Revision:**
> > >
> > >    - **Discussion on Full State Information Access During Training:**
> > >      Insert at line 106 of the related works section:
> > >      > "While it may appear idealistic to assume access to full state information during training, asymmetric RL for POMDPs [1-2] and RL for hindsight observable MDPs [3] have employed this assumption effectively to address various challenges, including self-driving vehicles [1]."
> > >
> > >    - **Discussion on Masked Observation Functions:**
> > >      Insert at the end of the related works section:
> > >      > "Masking is a practical representation of partial observability in real-world scenarios, such as robotics, where sensor occlusions can be interpreted as feature masks. Traditional POMDPs [3-6] typically employ a consistent observation function, mapping full states to partial ones based on predetermined factors. Unlike these approaches, our method introduces variability, as it avoids making specific assumptions about which factors will be masked during testing, leading to more dynamic and unpredictable mapping functions during evaluation."
> > >
> > >    - **Discussion on Awareness of Missing Vectors:**
> > >      Insert at the end of the related works section:
> > >      > "Our assumption about awareness of missing vectors, while not universally true, is pragmatically valid in many contexts, including robotics and network systems where missing or anomalous data can often be detected through sensor disturbances or specific anomaly indicators. This assumption is consistent with established practices in POMDP research [4] and studies on data missingness [7-8]."
> > >
> > > ---
> > > Thank you for your valuable comment again. If you have any other questions, please post them. We are happy to continue our communication.
> > >
> > > Authors

---

> > > ### Author Response · Authors · 2023-08-19
> > >
> > > Dear Reviewer AAho,
> > >
> > > Thank you once again for your invaluable feedback. We are approaching our discussion deadline soon, and have carefully considered your comments to provide further responses for your second feedback. Specifically, we are addressing the two main points you highlighted:
> > >
> > > - __Extensibility of Our Problem Setting Using Mask Observation Functions__:
> > >    We wish to emphasize that our framework is not confined to **explicit** mask observation functions. It is crafted to be adaptable to a variety of scenarios where parts of the state information are unobserved. Importantly, in our revised manuscript, we present several **concrete examples** that align well with our masked observation setting, demonstrating its broad applicability beyond traditional contexts in POMDP literature.
> > >
> > > - __Ensuring the Work's Scope is Evident from the Start__:
> > >    We have outlined specific revisions we intend to make to clearly convey the scope of our work right from the beginning. These revisions encompass the abstract, introduction, and related works sections, as detailed earlier.
> > >
> > > In light of the upcoming discussion deadline, we sincerely appreciate your time in reviewing our work. We would be most grateful for your prompt feedback on our responses, as this will greatly assist us in adhering to the discussion timeline. Please do not hesitate to let us know if you have any additional questions or require further clarifications. We eagerly look forward to your continued feedback and hope to engage in productive dialogue swiftly.
> > >
> > > Warm regards,
> > >
> > > Authors

---

> > > > ### Comment · Reviewer_AAho · 2023-08-21
> > > > **Reply**
> > > >
> > > > Thank you for the response. While changing the wording to _masked_ partial observability is better in terms of reducing scope, the authors have seemed to missed the point as to where issues lie.
> > > >
> > > > My main issue with the problem setting is _random masks_, not all masked observations. I agree wholly that masked observations is quite a common framework. The issue is with assuming a _random_ mask. This implies that for two given trajectories with the same underlying states, if $s_t^0$ is masked in the first trajectory, then there's a (pretty good) chance $s_t^1$ is not masked in the second trajectory. I would argue that the cases you've provided in your response are much harder versions of partial observability, because if you deterministically mask a state variable through _all_ trajectories, your task of inference of this hidden state variable becomes much harder. In addition to this, you assume the availability of an offline datasets with full state information during training. Observation functions (state to observation mappings) are at the crux of partial observability, and a simple random Bernoulli mask over states is one of the most simple cases.
> > > >
> > > > Since this main concern is still unaddressed, I am unchanged in my score.

---

> > > > ### Author Response · Authors · 2023-08-21
> > > >
> > > > Dear Reviewer,
> > > >
> > > > Thank you for your feedback. It appears there's a misunderstanding about our experimental setup, which we'd like to clarify.
> > > >
> > > > 1. **Types of Random Masking**: Our approach is not limited to using a simple random Bernoulli mask. We consider two distinct masking scenarios:
> > > >
> > > >     a. **Dynamical Missing Scenario** (Lines 235-237): This aligns with your interpretation of 'random masking,' where mask variables change dynamically. Results for this case are reported on lines 296-313 and Figure 4.
> > > >
> > > >     b. **Factor Reduction Scenario** (Lines 238-241): We believe this scenario directly addresses your concern regarding the deterministic masking of state variables. Here, mask variables are consistently set to simulate the removal of sensors, leading to a continuous lack of specific observations throughout each episode.
> > > >
> > > >     - **Details on Factor Reduction Scenario Testing**: In this scenario, we sample \(N\) sets of masked state variables, each corresponding to a different partial observation function. During testing, for each set of partial observations, we apply deterministic masking to the selected state variables across all trajectories. Performance metrics are then computed for each of these \(N\) sets by averaging the results across all trajectories. Despite the complexity this adds, our model has shown robust performance (lines 314-324, Figure 5), which suggests it is capable of meeting the challenges you've outlined.
> > > >
> > > > 2. **Training Assumptions and Efficacy**: Furthermore, during the training phase, our assumption of access to full observations is not novel; it has been consistently adopted in prior research. Given this assumption, we're afforded the flexibility to choose any masking strategy to devise masked partial observations. Of these strategies, random masking emerges as the most straightforward and effective choice. This efficacy of random masking has been validated not only in our experiments but also in several external studies across domains such as image processing [1] and RL tasks [2].
> > > >
> > > >
> > > >    [1] *Masked Autoencoders Are Scalable Vision Learners*, CVPR 2022
> > > >
> > > >    [2] *Masked Trajectory Models for Prediction, Representation, and Control*, ICML 2023
> > > >
> > > > Thank you for your valuable comment again. If you have any other questions, please post them. We are happy to continue our communication.
> > > >
> > > > Authors

---

> > > > > ### Comment · Reviewer_AAho · 2023-08-21
> > > > > **Response**
> > > > >
> > > > > Thanks for the prompt response.
> > > > >
> > > > > After a re-read the authors are correct; the factor reduction scenario covers observation masks in general. Re-evaluating the changes that the authors have suggested with regards to over-claiming, I will raise my score to a 4.
> > > > >
> > > > > As of now, I am not comfortable with raising my score any higher. My main quip with this work still holds: the mixing of the problem setting and solution method. Giving the ground-truth state during training is still a huge assumption, and is not present in the problem settings of all the examples the authors have listed. While there are works that do make this assumption, the current motivation for this solution method seems weak. Now that the boundaries of the problem setting are a bit more clear, the setting of hindsight state information comes to mind. This work might do well as a reformulation of this problem setting.

---

> > > > > > ### Author Response · Authors · 2023-08-22
> > > > > >
> > > > > > Thanks for your feedback, here is our clarifications:
> > > > > >
> > > > > > 1. **Assumption of Access to Full States During Training:** Many recent works have made this assumption, categorizable across three major areas:
> > > > > >
> > > > > >    - **Asymmetric RL for POMDPs** [1-4].
> > > > > >    - **Centralized Training and Decentralized Execution (CTDE) for MARL** [5-7].
> > > > > >    - **Hindsight observable MDPs** [8].
> > > > > >
> > > > > >     Moreover, our offline setting represents __a softer assumption__ than most works in these areas. While they often assume real-time access to complete states, we only require access to an offline dataset, which is a more practical scenario in many real-world applications.
> > > > > >
> > > > > > 2. **Clarifying Real-world Motivations:** We appreciate the feedback and would like to further underscore the real-world significance of our assumptions. In practical scenarios, observation functions can often be volatile and are not uniform across situations. While we have made certain assumptions for experimental clarity, they are fundamentally based on tangible real-world challenges.
> > > > > >
> > > > > >    a. **Search and Rescue Operations:**
> > > > > >    - **Preliminary Training:** A robot is often trained in controlled environments with access to comprehensive offline data. However, its real-world deployment can be substantially different from these conditions.
> > > > > >    - **Varied Challenges:** When dispatched to disaster zones, the robot can encounter unpredictable factors such as thick smoke, muddy waters, broken sensors, or electrical interference. Such elements effectively create dynamically masked observation functions — some sensors work some of the time, others might never work.
> > > > > >    - **Our Solution's Relevance:** In these critical situations, it's essential that the robot can still function effectively even if its sensory input is sporadic or compromised. Our approach aims to ensure that the model remains resilient and adaptable to these unforeseen changes in observation functions.
> > > > > >
> > > > > >    b. **Financial Markets:**
> > > > > >    - **Data-Driven Training:** Financial algorithms often get trained on comprehensive historical datasets. Yet, the real-world market is more volatile than any historical dataset can predict.
> > > > > >    - **Dynamic Market Conditions:** In ever-evolving markets, only certain subsets of indicators may be reliable at any given moment. Factors like geopolitical events, sudden economic shifts, or even technical glitches can dynamically mask some indicators while highlighting others.
> > > > > >    - **Our Solution's Relevance:** Financial losses due to misreading the market could be catastrophic. Our methodology prepares algorithms to cope with these dynamic observation functions, making financial predictions more resilient to real-world market volatilities.
> > > > > >
> > > > > > 3. **Differentiation from Hindsight MDP:** It's crucial to highlight that while our work might share some similarities with the Hindsight state information setting, significant differences exist:
> > > > > >
> > > > > >    __Table 1: Comparing our method to the Hindsight state information setting__
> > > > > >
> > > > > > | Method | Access during training | Mode | Observation function |
> > > > > > | :-----: | :---------------------: | :---: | :-------------------: |
> > > > > > | Ours | Yes (Offline Dataset) | Offline | Diverse and uncertain mask observation functions |
> > > > > > | Hindsight state information | Yes | Online | Singular, specific observation function |
> > > > > >
> > > > > > [1] Robust asymmetric learning in pomdps. ICML 2021.
> > > > > >
> > > > > > [2] Leveraging fully observable policies for learning under partial observability. CoRL 2022.
> > > > > >
> > > > > > [3] Bridging the imitation gap by adaptive insubordination. NeurIPS 2022.
> > > > > >
> > > > > > [4] Deep q-learning from demonstrations. AAAI 2018.
> > > > > >
> > > > > > [5] QMIX: Monotonic Value Function Factorisation for Deep Multi-Agent Reinforcement Learning. ICML 2018.
> > > > > >
> > > > > > [6] The Surprising Effectiveness of PPO in Cooperative, Multi-Agent Games. NeurIPS 2022.
> > > > > >
> > > > > > [7] Online Ad Hoc Teamwork under Partial Observability. ICLR 2022.
> > > > > >
> > > > > > [8] Learning in POMDPs is Sample-Efficient with Hindsight Observability. ICML 2023.

---

### Official Review · Reviewer_iLBR · 2023-07-06

**Soundness:** 3 good
**Presentation:** 3 good
**Contribution:** 3 good
**Rating:** 6
**Confidence:** 3

**Summary:**

This work study the masked partial observability in reinforcement learning and propose a novel method ORDER to address this challenge. ORDER leverages the alignment of discrete state representations and significantly improves robustness and generalization performance across diverse masked partial observability. Experiments demonstrate the effectiveness of ORDER in some settings.

**Strengths:**

This work leverages the alignment of discrete state representations learned by the fully observable offline data and discrete proxy representations for the partial observable experiences, novelly constructing a proxy policy for the partial observable online interaction, which improves the robustness and generalization for the diverse partial observation.

**Weaknesses:**

The author claims that ORDER is suitable for heterogeneous state factors, but only provides the experiment results for simple homogeneous state factors settings.

**Questions:**

1. Whether it is reasonable to convert any unobservable state factor into a single mask token $e^{[mask]}$? Is there any convincing reason or it is just a practical implementation to enable the algorithm execution? $e^{[mask]}$ is learned by minimizing (6) along with the tray encoder and prediction heads, can you explain what it means to learn such a mask token?
2. You mentioned that the state factors can be heterogeneous, but it seems that your experiments only include homogeneous cases where each dimension of the observation vector is treated as an independent state factor. I can accept it as a simplified experiment setting, but I wonder that in general heterogeneous cases where each state factor differs in type and size, how to choose the numbers of discrete codes for each factor? A common discretization degree seems unreasonable now.
3. Do some kind of heterogeneous state factors like high-dimensional multimedia data suitable for discrete representations? Since in your experiments, each state factor is a single scalar, which is an extreme simplification compared to the heterogeneous factors that you claimed, I really doubt this point.

**Limitations:**

ORDER currently focuses on masked partial observation, leaving other forms like perturbations and noises unaddressed. Moreover, its applicability to complex heterogeneous high-dimensional real-world tasks remains to be explored.

---

> ### Author Rebuttal · Authors · 2023-08-10
>
> Dear Reviewer:
>
> Thank you for your thoughtful questions and feedback on our paper. Below, we provide detailed responses to each of your questions:
>
> # Single Mask Token Justification
>
> Thanks for your feedback. Your query regarding the use of a single mask token is apt. The primary intention behind using just one mask token is to provide the network with a cue that certain information is occluded, rather than detailing the specific content of the occlusion. This serves as an efficient indicator. In practice, this approach finds resonance with prevailing techniques in transformer architectures, such as in the MAE [1] model. The essence is to highlight the presence of an occlusion rather than its intricacies, for which a single token suffices.
>
> [1] *Masked Autoencoders Are Scalable Vision Learners.* CVPR 2021.
>
> # Addressing Heterogeneity in State Factors
>
> Thanks for your feedback. Your observation regarding the treatment of state factors in our experiments is keen. In the mujoco tasks that we worked on, state factors are indeed heterogeneous in nature. Specifically, the state vector amalgamates two distinct types of data: the robot's sensor positions and its angular velocities.
>
> When state factors diverge in type and magnitude, a reasonable approach to select the number of discrete codes would be to gauge the redundancy of information inherent to each state factor. In simpler terms, state factors carrying more intricate or effective information might necessitate a greater number of discrete codes. To facilitate this operation, expertise from human professionals can be sought or alternative information measurement techniques can be employed. This ensures a tailored discretization that aligns with the unique characteristics of each state factor.
>
> # Addressing High-dimensional Multimedia Data
>
> Thank you for your thoughtful comment.  We agree that this is an important consideration. In support of our approach, we refer to a relevant study [1] in the domain of model-based reinforcement learning. This work demonstrates the efficacy of discrete representations in enhancing policy performance, particularly when dealing with high-dimensional data such as images in gaming scenarios. Notably, the findings indicate that encoding such data into discrete representations not only preserves performance but also enhances robustness by mitigating the impact of information noise associated with high-dimensional factors.
>
> We acknowledge the potential oversimplification in our current experimental setup and will address this concern by including a concise discussion in the related work section, highlighting the relevancy of discrete representations for handling high-dimensional multimedia data.
>
>
> [1] *Mastering Atari with Discrete World Models.* ICLR 2021.

---

> > ### Comment · Reviewer_iLBR · 2023-08-22
> >
> > Thank you for your response. I think this paper to be generally well-done. However, it requires a more specific design approach when dealing with heterogeneous state factors. Therefore, I will maintain my current score.

---

> > > ### Author Response · Authors · 2023-08-22
> > >
> > > Dear Reviewer,
> > >
> > > Thank you for recognizing the merits of our paper and offering valuable feedback.
> > >
> > > Regarding your suggestion on a more specific design approach for heterogeneous state factors, we appreciate the insight. Indeed, different state encoders can be tailored for diverse data modalities. For instance:
> > > - **Image Data:** We could employ a CNN to effectively capture image-related state information.
> > > - **Vector-Based Data:** An MLP can be optimized to address vector-based state factors.
> > > - **Text-Based Data:** For state information present in textual format, transformers can be an ideal choice.
> > >
> > > Furthermore, by adjusting the hyperparameters related to the codebook dimension and discrete codes count, we can finely tune the information capacity for each heterogeneous state factor. This approach allows for a more nuanced and effective representation of diverse state information.
> > >
> > > We hope this clarifies our approach and assuages any concerns you might have.
> > >
> > > Warm regards,
> > >
> > > Authors

---

### Official Review · Reviewer_iZED · 2023-07-06

**Soundness:** 2 fair
**Presentation:** 3 good
**Contribution:** 3 good
**Rating:** 7
**Confidence:** 3

**Summary:**

In this paper, the authors present a three stage method for offline RL in POMDPs. It is assumed that the agent has access to full observations (state) during training, but only masked versions of this state (partial observations) during inference or deployment. In the first stage of training, a mapping is learnt from the original state space to a discrete state space. In the second stage, an optimal policy is learnt using offline RL which maps this discrete state to an action. Finally, in the third stage, a proxy state representation model is learnt, which maps the history of partial observations to a proxy discrete state, which is aligned with the discrete state corresponding to the underlying system state. As different state factors can be masked during inference, this method presents a solution that generalises well to a family of POMDPs obtained from the underlying MDP using different masking strategies. The authors demonstrate this method on four MuJoCo environments.


**Strengths:**

1. The paper is novel, very well written and easy to comprehend.
2. The paper presents generalising an RL solution of an MDP to a family of POMDPs, thereby making RL generalisable and fairly robust to several real-world problems with a type of partial observations (masked state factors).

**Weaknesses:**

1. There is no analysis/theory presented regarding loss of optimality due to the discrete state representation.


**Questions:**

1. [Typo] In Line 164, shouldn't the serial number be "i" instead of "ii"?
2. [Typo] In Eq(1) is a term missing after "where"?
3. How can it be proven that the discrete state obtained is indeed a state sufficient for control?
4. I did not understand the difference and the motivation between the second and the third terms in Eq (3). Can the authors please explain the same?
5. \tau is not defined. Also, the subscript corresponding to \tau is sometimes just "t-1" and sometimes "0:t-1". Can these be made consistent with a proper definition?
6. [Typo] In line 230 consider replacing "has" with "have".
7. [Typo] Reference missing (??) in Line 66 of Supplementary Material.

**Limitations:**

The authors have covered the limitations of their method. While they do not mention the societal impact, since the paper deals with a general algorithm, the societal impact is same as in any RL case.

---

> ### Author Rebuttal · Authors · 2023-08-10
>
> Dear Reviewer:
>
> Thank you for your thoughtful questions and feedback on our paper. Below, we provide detailed responses to each of your questions:
>
> # Response to Typographical Comments:
>
>    1. **Line 164**: You're right, and we apologize for the oversight. It should indeed be "i" instead of "ii". This will be rectified.
>
>    2. **Eq(1)**: Thank you for pointing it out. The term $ j $ was indeed missing after "where". We will correct this in the revised manuscript.
>
>    3. **Line 230**: Your observation is accurate, and the correction has already been implemented.
>
>    4. **Line 66 of Supplementary Material**: We apologize for the omission. The intended reference is "Algorithm 4". We'll ensure it is properly cited in the revision.
>
> # Response to the Discrete State Query:
>
> Your query regarding the sufficiency of the discrete state for control is insightful. To demonstrate the sufficiency of our discrete representation:
>
>    1. **Performance Metrics**: We utilize standard benchmarks and performance metrics in our experiments. The fact that our method, using the discrete state representation, achieves competitive or superior performance indicates that the representation captures the necessary state information for effective control.
>
>    2. **Robustness in Varied Environments**: In addition to standard benchmarks, we evaluate our approach in a variety of environments and scenarios. The discrete state's ability to generalize across these settings further attests to its sufficiency.
>
>    3. **Comparison with Continuous Representation**: We juxtapose the performance of our discrete representation with its continuous counterpart. The discrete representation's comparable or better performance suggests its capability to serve as an efficient and sufficient control state.
>
> In the future, one could also explore methods to directly measure the information content or fidelity of the discrete state against the true continuous state. However, for the scope of this paper, our empirical results act as a testament to the discrete state's efficacy for control tasks.
>
> # Response to Query on Eq (3) Terms:
>
> Your query regarding the second and third terms in Eq (3) is pertinent. To provide clarity:
>
>    1. **Purpose of Both Terms**: Both terms originate from the VQ-VAE framework, aimed at ensuring high-quality representations during training.
>
>    2. **Difference in Position of  $sg()$**: The positioning of the "stop gradient" function, $sg()$, differentiates the terms. It's pivotal in dictating where the gradient propagates during backpropagation.
>
>    3. **Separate Control Mechanisms**: The distinct placements of $sg()$:
>        - The second term, with $sg()$, focuses on updating the codebook, ensuring it reflects the encodings accurately.
>        - The third term, conversely, guides the encoder learning so that its outputs align closely with the existing codebook entries.
>
>    4. **Intuitive Takeaway**: Think of it as a dance between two partners, the encoder and the codebook. The second term ensures the codebook learns to match the encoder's steps (representations). Simultaneously, the third term ensures the encoder does not stray too far from what the codebook knows, offering a consistent dance.
>
> By carefully balancing these terms using scale coefficients, we can achieve a harmonious alignment between the encoder and the codebook, ensuring optimal representations.
>
> # Response to Query on $\tau$ Definition:
>
> Thank you for pointing out the inconsistency regarding $\tau$. We apologize for the confusion.
>
> 1. **Definition of $\tau$**: As highlighted, $tau$ is indeed defined in lines 122-123 of the manuscript.
>
> 2. **On Subscript Clarification**: We understand the confusion and regret the inconsistency. To clarify, the notation "$\tau_{t-1}$" is a typographical error. The correct notation is "$\tau_{0:t-1}$", which denotes the entire trajectory from the start up to "t-1".
>
> We will rectify the inconsistency throughout the manuscript to ensure "$\tau_{0:t-1}$" is used consistently and appropriately. Your feedback is instrumental in enhancing the clarity of our work.

---

> > ### Comment · Reviewer_iZED · 2023-08-19
> >
> > I thank the authors for incorporating some of my suggestions and also addressing my questions. I am still not clear how the discrete state representation is a sufficient statistic or an information state. The argument that this representation empirically yields better results does not necessarily entail good performance in other environments. I thank the authors on the clarification for the various terms in Eq(3). This helped me in understanding the equation.

---

> > > ### Author Response · Authors · 2023-08-20
> > >
> > > Dear Reviewer,
> > >
> > > Thank you once again for your invaluable feedback. To address the question of sufficiency of the discrete state for control, we rely on a two-pronged approach:
> > >
> > > 1. **Reconstruction Capability**: The loss function in Equation (3) contains a reconstruction term as its first component. This term is designed to maximize the information encapsulated in the discrete state so that it can adequately reconstruct the original state. In essence, if the discrete state can accurately reconstruct the original state, it carries enough information to be useful for control tasks.
> > >
> > > 2. **Hyperparameter Tuning**: We acknowledge that discrete representations can lose some information compared to the original states. To counteract this and ensure the discrete states are sufficient for control, we offer flexibility in the model through hyperparameters. Specifically, users can adjust the number of discrete codes and the dimension of the codebook. This fine-tuning enables the model to capture the necessary amount of information to maintain control quality.
> > >
> > > Through these mechanisms, we aim to ensure that the discrete states generated are indeed sufficient for effective control.

---

> > > > ### Comment · Reviewer_iZED · 2023-08-20
> > > >
> > > > Thanks for this clarification regarding the discrete state as a sufficient statistic.

---

> > > > > ### Author Response · Authors · 2023-08-20
> > > > >
> > > > > Dear Reviewer,
> > > > >
> > > > > Thanks for your response again. We appreciate your valuable feedback on improving our work. If you have any other questions, please post them and we are happy to continue our communication.
> > > > >
> > > > > Best Regards,
> > > > > Authors

---

### Author Rebuttal · Authors · 2023-08-10

**Response to Reviewers:**

Dear Reviewers,

Thank you for your comprehensive feedback on our manuscript. We have addressed each comment individually in the subsequent sections. Additionally, an attached PDF provides further clarifications with relevant tables.

Your insights are crucial for refining our work, and we trust our responses and modifications align with your suggestions.

Looking forward to your continued feedback.

Best regards,

Authors

---

### Decision · Program_Chairs · 2023-09-21

**Decision:**

Accept (poster)

**Comment:**

The overall assessment is that the paper made a solid contribution. The primary debate was about the generality of the problem being considered and how well it mapped to real-world tasks. This is certainly a point that can be discussed more thoroughly in the paper. However, that potential weakness is well balanced by the interesting approach and strong experimental results.

Please take the critical feedback into account when preparing the camera ready paper.